biomathematics

hyperthermia, perfusion, blood flow, mathematical model

**Author for correspondence:**
Jesús J. Bosque
e-mail: jesus.bosque@uclm.es

# The interplay of blood flow and temperature in regional hyperthermia: a mathematical approach

Jesús J. Bosque[1], Gabriel F. Calvo[1],
Víctor M. Pérez-García[1] and María Cruz Navarro[2]

[1]Department of Mathematics, Mathematical Oncology Laboratory (MOLAB), and [2]Department of Mathematics-IMACI, Facultad de Ciencias y Tecnologías Químicas, University of Castilla-La Mancha, Ciudad Real, Spain

(iD) JJB, 0000-0002-9116-1453

In recent decades, hyperthermia has been used to raise oxygenation levels in tumours undergoing other therapeutic modalities, of which radiotherapy is the most prominent one. It has been hypothesized that oxygenation increases would come from improved blood flow associated with vasodilation. However, no test has determined whether this is a relevant assumption or other mechanisms might be acting. Additionally, since hyperthermia and radiotherapy are not usually co-administered, the crucial question arises as to how temperature and perfusion in tumours will change during and after hyperthermia. Overall, it would seem necessary to find a research framework that clarifies the current knowledge, delimits the scope of the different effects and guides future research. Here, we propose a simple mathematical model to account for temperature and perfusion dynamics in brain tumours subjected to regional hyperthermia. Our results indicate that tumours in well-perfused organs like the brain might only reach therapeutic temperatures if their vasculature is highly disrupted. Furthermore, the characteristic times of return to normal temperature levels are markedly shorter than those required to deliver adjuvant radiotherapy. According to this, a mechanistic coupling of perfusion and temperature would not explain any major oxygenation boost in brain tumours immediately after hyperthermia.

## 1. Introduction

Hyperthermia treatment (HT) in cancer has been regarded as an adjuvant treatment to improve the efficacy of other traditional

therapies, most often radiation therapy [1,2]. A number of preclinical studies have been carried out showing the benefits of using radiotherapy on target tissues that also receive thermal doses [3]. In addition, various randomized clinical trials on different tumours have shown partial/total remissions [4,5]. This radio-sensitization is attributed to several biological mechanisms and interactions [6], such as direct cell killing by heat [7], DNA repair inhibition [8], short-term [9] and long-term improved oxygenation [10,11], protein denaturation [12], preferential vascular damage [13] or other effects like immune system activation [14]. Despite the long history of this therapeutic modality, currently there is no conclusive data making it possible to determine which of these HT-induced mechanisms has the most substantial impact on treatment response.

Blood flow is widely recognized as being influenced by heat and plays a twofold role in hyperthermia. Firstly, it is a key factor in heat dissipation within a tissue [15]. Secondly, it has the biological function of delivering nutrients and oxygen, the latter being of great importance when hyperthermia is used in combination with radiotherapy, as oxygenated tissues display a better response to radiotherapy [16]. Blood flow acts as a convective sink for heat in tissues and thus it may greatly affect the temperature of tumour regions during HT. In addition, it has been frequently reported that a local higher temperature leads to an increase in the blood flow irrigating the tissue [17]. Thus, temperature and blood flow are two mutually interacting quantities that have to be considered in a unified way. Hence, increases in perfusion, prompted by higher temperatures, are expected to improve oxygenation and, consequently, the response to radiation therapy. However, in the clinic, hyperthermia and radiotherapy are not administered simultaneously [18]; depending on the specific institution they are sequentially applied in a different order and with varying time intervals. Therefore, the important and practical problem emerges of determining how temperature and perfusion will evolve both during and after treatment, according to their mutual mechanistic association, and also how an elevation in perfusion could impact on a sustained enhanced oxygenation after hyperthermia is delivered.

Here, we address the use of regional hyperthermia in brain tumours. Malignancies of the central nervous system are rare, representing about 1.5% of new cancer diagnoses in 2020, but they make up over 3% of all cancer-related deaths [19], and are a significant cause of morbidity and mortality in children and young adults [20]. Due to the challenges that exist in treating these malignancies, particularly in high-grade gliomas, the possibility of taking advantage of elevated temperatures to improve the outcome of standard therapies is frequently used in the clinic. For instance, previous clinical trials with 79 randomized glioblastoma patients showed a significant benefit, both in time to progression and overall survival, for the group receiving hyperthermia plus radiotherapy when compared with the group that only received radiotherapy [21]. In that study, heat was administered via an invasive intracranial device consisting of interstitial antennas, both before and after the application of brachytherapy. Currently, a number of different and less invasive procedures are employed in the clinic [22], but there is still great interest in how heat can improve the outcomes of brain tumour patients.

Mathematical models have been proposed as a way to gain insight into the hallmarks of cancer and the therapeutic approaches [23,24]. Specifically, in hyperthermia, the process of heating tissues and the interaction with other therapies has received attention from the mathematical standpoint [25,26]. Mechanistic models, with varying levels of complexity, have been developed aimed at calculating temperatures [27] and, less frequently, to quantify the relevance of the biological effects intervening during treatment [28]. In this study, we put forward a lumped parameter mathematical model based on energy conservation laws that captures the essential features of the interplay between temperature and blood perfusion in heated tumour tissue. We focus our study on the coupling between both variables, and evaluate their dynamics during and after treatment in brain tumours with the goal of identifying limitations in current procedures as well as pinpointing possible room for improvement in the clinical setting.

# 2. Methods

## 2.1. Model formulation

To model the temperature evolution in a tumour subjected to regional hyperthermia and the variations in the blood flow in response to the administration of heat, we use the law of conservation of energy, accounting for three separate macroscopic compartments which exchange heat through conduction and convection (figure 1a,c): tumour tissue (no subscript for variables), healthy tissue surrounding the tumour (subscript s) and blood feeding both types of tissues (subscript b). Each compartment will be

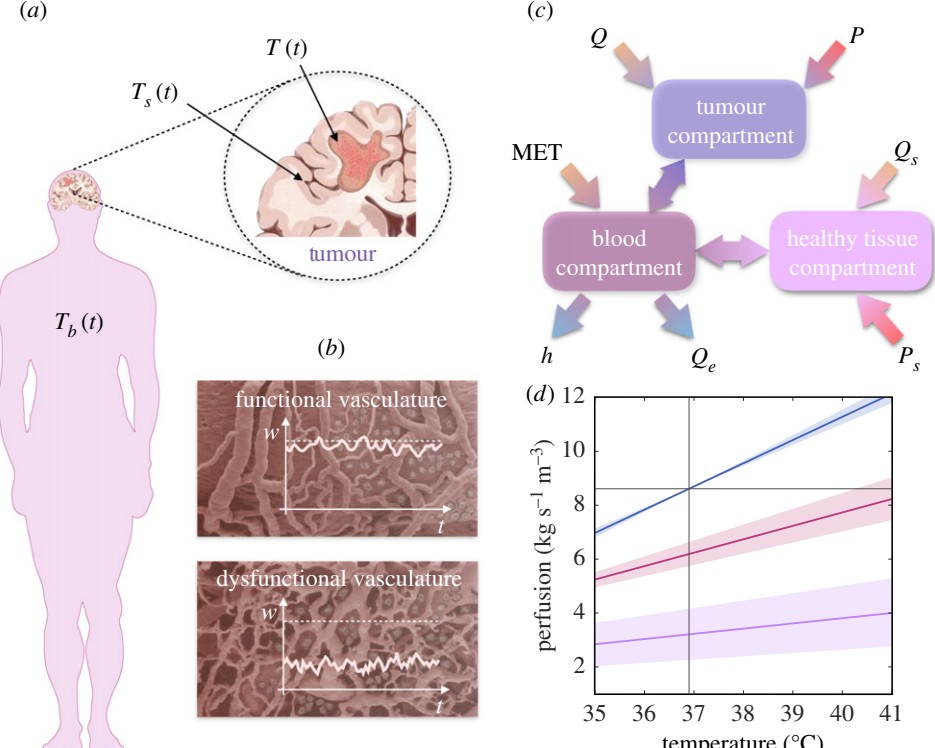

**Figure 1.** Graphical representation of the mathematical model for temperature and perfusion evolution during hyperthermia treatment. (a) The human silhouette shows the distinct body regions related to each compartment. The tumour is considered to be located within the brain (for instance, a glioma), and corresponds to the first compartment, while the healthy brain parenchyma (surrounding tissue), is the second. The blood feeding both the brain and the tumour, which circulates through the rest of the body, exchanging heat with it and with the environment, constitutes the third compartment. (b) The response of the compartments to heat will be highly dependent on the functionality of their vasculature. Malignant tumours are characterized by aberrant blood vessels and different tumours will respond disparately to heat according to the functionality of their vasculature, which translates into starkly dissimilar behaviour over time of the associated perfusion w. (c) The mathematical model given by equations (2.1) incorporates the energy relations represented and aims at providing the temperature and blood flow variation in the three compartments during heat administration and subsequently. (d) Linear dependence between blood flow and temperature for different tissues considered in our mathematical model. The colour code is consistent with the subsequent figures. The blue line represents the healthy tissue whereas the pink and purple lines represent two tumours with distinct vasculature functionality status. For impaired vasculature, the uncertainty associated with the tumour perfusion increases, while its heat response capacity (the slope of the line) is lower.

assigned to a single representative average value of the temperature ($T$, $T_s$, $T_b$ [°C]) and the compartments for the tumour and healthy tissues will be characterized by a single value for the baseline blood perfusion within each tissue ($w$, $w_b$ [kg s$^{-1}$ m$^{-3}$], though usually reported as ml of blood per 100 g of tissue per minute).

The first compartment (tumour) has a characteristic density $\rho$ [kg m$^{-3}$] and a heat capacity $c$ [J °C$^{-1}$ kg$^{-1}$]. It receives energy from the external source, which induces an internal effective power density $P$ [W m$^{-3}$], and by its own metabolic heat production $Q$ [W m$^{-3}$]. The tumour exchanges energy with the blood entering through the arteries with a temperature $T_b$ [°C] at a rate $w$ [kg s$^{-1}$ m$^{-3}$]. This represents a contribution of $wc_bT_b$ [W m$^{-3}$], where $c_b$ [J °C$^{-1}$ kg$^{-1}$] is the heat capacity of the blood. We follow Pennes' assumption that blood leaves tissue through the veins at the tissue temperature $T$ [29], explained by the large heat exchange surface between tumour and blood in the vessel network. Even though Pennes' framework has been criticized (e.g. [30]), it is important to note that our assumptions do not hold locally at every point of the tissue, as his equation does, but rather it comprises all possible aggregate macroscopic contributions for the tissue as an energy rate balance for a control volume.

The second compartment, corresponding to the healthy tissue surrounding the tumour (mass density $\rho_s$, heat capacity $c_s$), shares the same modelling scheme as the previous one, being heated by an internal metabolic rate of $Q_s$ [W m$^{-3}$] and by an external radio-frequency source that produces an effective internal power density $P_s$ [W m$^{-3}$] that could be different from that of the tumour tissue. Healthy

tissue exchanges heat with blood, yielding a term $w_s c_b(T_b - T_s)$ [W m$^{-3}$] for the gross energy gain. The first and second compartments would also exchange heat, by thermal conduction, both with each other and with the environment, and diffusive terms could be included to account for this process. However, those contributions are neglected here as previous works have shown that perfusion is the main process in heat transfer within the brain and conduction is only important in a limited region of only a few millimetres [31,32]. Furthermore, since the healthy tissue is part of an internal organ, the brain, all the losses to the environment are accounted for via the blood that carries the heat to the skin. The mathematical description of other organs with a larger interface with the environment, as would be the case of the breast, may require those contributions to be accounted for.

The third compartment (blood) receives the heat that the other two have exchanged according to the amount of flow reaching the first and second compartments proportionally to their respective volumes, $V$ and $V_s$. The total mass affected by the heat exchange in this compartment is the total body mass $m_T$ [kg] minus the mass of the first and second compartments already considered, $\rho V$ and $\rho_s V_s$, respectively. The heat capacity in this case is the effective one for the body $c_T$ [J °C$^{-1}$ kg$^{-1}$]. This compartment is also fed by the metabolic body heat MET, excluding what has already been attributed to the other compartments, $QV$ and $Q_s V_s$ [W]. Lastly, this compartment accounts for the overall losses of thermal energy to the environment, which, according to Newton's Law, depends on the difference between its average temperature and that of the environment $T_E$ [°C], the surface of the body $A$ [m$^2$] and an effective heat transfer coefficient by the skin $h$ [W m$^{-2}$ °C$^{-1}$].

The aforementioned relationships among the three different compartments in a brain tumour are illustrated in figure 1c. These considerations are expressed by the following system of three ordinary differential equations (ODEs) for the temperatures in each compartment

$$\frac{dT}{dt} = \frac{1}{\rho c}[wc_b(T_b - T) + Q + P], \tag{2.1a}$$

$$\frac{dT_s}{dt} = \frac{1}{\rho_s c_s}[w_s c_b(T_b - T_s) + Q_s + P_s], \tag{2.1b}$$

$$\frac{dT_b}{dt} = \frac{1}{m_T - \rho V - \rho_s V_s}\frac{1}{c_T}[Vwc_b(T - T_b) + V_s w_s c_b(T_s - T_b) + \text{MET} - QV - Q_s V_s$$
$$- hA(T_b - T_E) - Q_e]. \tag{2.1c}$$

System (2.1) represents a lumped parameter model which provides spatially averaged temperature dynamics in the three main compartments considered.

## 2.2. Blood perfusion

The perfusion terms $w$, $w_s$ play an essential role in equations (2.1). They are assumed to display a moderate dependence on temperature, thus making 2.1 a nonlinear ODE system. Moriyama studied the changes in cerebral blood flow (CBF) in monkeys exposed to interstitial microwave hyperthermia [33]. The author found that in those animals the CBF increased with temperature at a rate of 10% for each degree increase in temperature. Normal flow levels were restored after concluding heat administration. Rosomoff studied the variation in CBF during hypothermia in dogs and found a proportional decrease in temperature variation of around 7% per degree [34]. Overall, these studies suggest that the variation in CBF is proportional to the temperature change and that the functional dependence of that change is somewhat similar whether the variation is upward or downward. Based on those studies, we propose a change in perfusion rates proportional to the temperature with a linear variation of the characteristic basal perfusion $w_0$ (in units [kg s$^{-1}$ m$^{-3}$]) of the form

$$w = w_0 \zeta[1 + \gamma(T - T_{\text{eq}})] \tag{2.2a}$$

and

$$w_s = w_0[1 + \chi(T_s - T_{\text{seq}})], \tag{2.2b}$$

where $T_{\text{eq}}$ and $T_{\text{seq}}$ refer to the homeostatic equilibrium values of the temperature in the tumour and its surrounding tissue, respectively, whereas the parameters $\gamma$ and $\chi$ (in units [°C$^{-1}$]) are the proportionality constants of these linear dependencies. Furthermore, an additional parameter $\zeta$ has been introduced to reflect the vascular impairment status within the tumour region, which is expected to display, on average, a different net blood flow in comparison to the surrounding healthy tissue. The functional

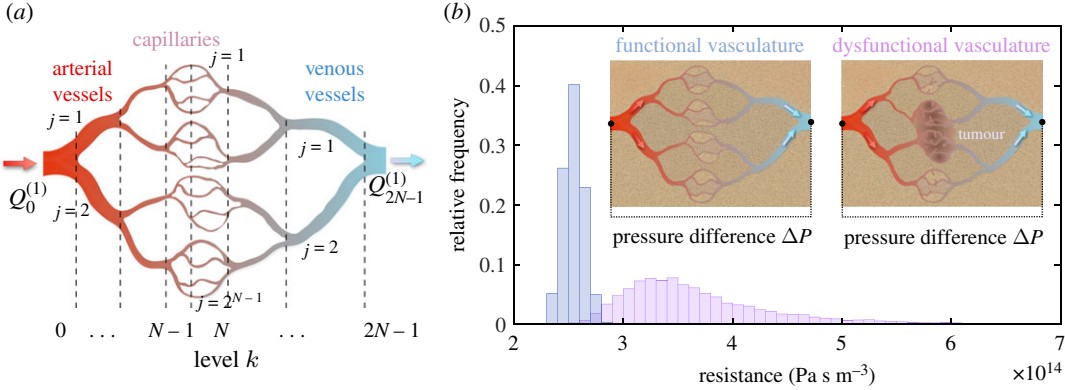

**Figure 2.** (a) Tree-like vascular network in which parent arterial vessels bifurcate into two daughter branches down to the capillary levels which then merge to form venous vessels. The networks are constructed starting from the capillary levels with characteristic radii and lengths of $r_{N-1}^{(c)} = 5$ μm and $l_{N-1}^{(c)} = 75$ μm and recursively applying well-known scaling laws for the previous levels. At each level, stochastic changes are allowed for both the radii $r_k^{(j)}$ and lengths $l_k^{(j)}$ of each vessel $(k, j)$, the variability being higher for dysfunctional (tumour) vasculatures. (b) Histograms for the total resistances (see the text) $Z_{tot}^{(nor)}$ and $Z_{tot}^{(tum)}$ in functional (normal) and a dysfunctional (tumour) vasculatures with $N = 6$ arterial levels (same number for venous levels). The pressure difference $\Delta P$ between some reference input arterial vessel and an output venous vessel in both networks is the same. The mean value of $Z_{tot}^{(nor)}$ is $2.547 \times 10^{14}$ Pa s m$^{-3}$, with a standard deviation of $0.091 \times 10^{14}$ Pa s m$^{-3}$, while the mean value for $Z_{tot}^{(tum)}$ is $3.744 \times 10^{14}$ Pa s m$^{-3}$, with a standard deviation of $0.862 \times 10^{14}$ Pa s m$^{-3}$.

relation of blood flow values to temperature is graphically illustrated in figure 1*d* for three tissues: the healthy brain and two abnormal tissues with disparate degrees of impairment and fluctuations. Such fluctuations incorporate temporal stochastic components into the blood flow of tumours, which are often characterized by a tortuous vasculature that cannot continuously satisfy the requirements imposed by the cells [35].

## 2.3. Vascular network model

In order to provide biological insight into the role of the phenomenological parameter $\zeta$ in our model, we will consider two scenarios (figure 2), each involving a simple vascular network: one functional and the other dysfunctional crossing a tumour region. Both networks will be assumed to be tree-like, sufficiently large and of similar size. Consequently, an equal reference pressure difference $\Delta P$, corresponding to the difference between a generic feeding artery and a draining vein, can be established so that the comparison in performance of the two networks is meaningful. Moreover, each parent arterial vessel will bifurcate into two daughter branches down to the capillary levels which will subsequently merge to form venous vessels. The number of arterial and venous levels, when either a branching or a merging takes place, will be $N$. Although the two networks considered will be relatively symmetric, we allow for the possibility of changes both in the radius $r_k^{(j)}$ and length $l_k^{(j)}$ of each vessel at level $k$, with $k = 0, 1, …, N-1$ for arterial vessels, and $k = N, N+1, …,$ $2N-1$ for venous vessels. At each level $k$, the index $j$ will run over the ranges $j = 1, 2, …, 2^k$ and $j = 1,$ $2, …, 2^{2N-1-k}$ for arterial and venous vessels, respectively. The blood flow through each network will be approximated as an ideal laminar flow obeying the Hagen–Poiseuille Law [36], where the flow rate $Q_k^{(j)}$ at each vessel, labelled by integers $(k, j)$, satisfies $Q_k^{(j)} = \Delta P_k^{(j)}/Z_k^{(j)}$, where $\Delta P_k^{(j)}$ is the corresponding pressure drop and $Z_k^{(j)}$ the resistance, given by $Z_k^{(j)} = 8\mu l_k^{(j)}/\pi(r_k^{(j)})^4$, where $\mu$ is the dynamic viscosity of the fluid.

If in the networks described we count over all possible paths along the $2N$ vessel levels, each path being associated with a total pressure drop $\Delta P$, we can write

$$\Delta P = \sum_{k=0}^{N-1} \sum_{j=1}^{2^k} \frac{Z_k^{(j)} Q_k^{(j)}}{2^k} + \sum_{k=N}^{2N-1} \sum_{j=1}^{2^{2N-1-k}} \frac{Z_k^{(j)} Q_k^{(j)}}{2^{2N-1-k}}, \tag{2.3}$$

where the first and second terms come from adding up the ongoing (arterial) and the return (venous) trees, respectively. Next, we impose conservation of blood flow at each junction point, i.e. $Q_k^{(j)} = Q_{k+1}^{(2j-1)} + Q_{k+1}^{(2j)}$ and $Q_k^{(2j-1)} + Q_k^{(2j)} = Q_{k+1}^{(j)}$, at arterial and venous sections, respectively. If we further assume that at each

bifurcation of the network, either at the arterial or venous sections, $Q_{k+1}^{(2j-1)} = Q_{k+1}^{(2j)}$ holds, then it follows that $Q_k^{(j)} = Q_0^{(1)}/2^k$, for $k = 0, 1, \ldots, N-1$ and $Q_k^{(j)} = Q_{2N-1}^{(1)}/2^{2N-1-k}$, for $k = N \ldots, 2N-1$, respectively. Here, $Q_0^{(1)}$ and $Q_0^{(2N-1)}$ denote the input and output flow rates in the network, and since the network is lossless, they are both equal, say, to $Q_0$. Hence, expression (2.3) can be approximated by

$$\Delta P = \left[ \sum_{k=0}^{N-1} \sum_{j=1}^{2^k} \frac{Z_k^{(j)}}{2^{2k}} + \sum_{k=N}^{2N-1} \sum_{j=1}^{2^{2N-1-k}} \frac{Z_k^{(j)}}{2^{4N-2-2k}} \right] Q_0 \equiv Z_{tot} Q_0, \tag{2.4}$$

where $Z_{tot}$ is the total resistance of the network.

Given the morphometric data of a specific vascular network, $Z_{tot}$ can be evaluated using the expression between square brackets in (2.4). However, for sufficiently large networks, the number of levels arising implies that the factors $2^k$ and $2^{2N-1-k}$ will be much greater than one. Hence, we may approximate the summation terms, involving the index $j$, by integrals and thus write

$$Z_{tot} \simeq \frac{8\mu}{\pi} \sum_{k=0}^{N-1} \int_{r_k^{(min)}}^{r_k^{(max)}} \int_{l_k^{(min)}}^{l_k^{(max)}} \frac{[f_k(r, l) + f_{2N-1-k}(r, l)]l}{2^k r^4} \, dr \, dl, \tag{2.5}$$

where $f_k(r, l)$ and $f_{2N-1-k}(r, l)$ denote the joint distribution functions of radii and lengths for arterial and venous vessels at levels $k$ and $2N - 1 - k$, respectively, whereas $r_k^{(min)}$, $r_k^{(max)}$, $l_k^{(min)}$ and $l_k^{(max)}$ represent the minimum and maximum radii and lengths of the vascular network at vessel level $k$. These distribution functions account for the fact that, in general, the radii and lengths of the vessels can be regarded as random variables fluctuating around average geometric values obeying intraspecific scaling laws of the vasculature [36].

The characteristic hierarchy found in normal vasculature, which is evenly spaced and displays well-differentiated arteries, arterioles, capillaries, venules and veins, becomes highly disorganized in tumour vessel networks [37]. In general, it is observed that the relation between radii and lengths in successive levels can be considered to strongly fluctuate with respect to the average geometric values encountered in the normal case. This implies that the corresponding joint distribution functions in (2.4) will show relatively larger deviations in the aberrant than in the normal vasculature. Thus, the total resistances $Z_{tot}^{(nor)}$ and $Z_{tot}^{(tum)}$ in normal and tumour vasculatures will be significantly different, with $Z_{tot}^{(tum)}$ expected to be greater than $Z_{tot}^{(nor)}$. Therefore, if $Q_0^{(nor)}$ and $Q_0^{(tum)}$ are the flow rates in normal and tumour vascular networks, and since by assumption both networks have an equal pressure difference $\Delta P$, the following relation will hold for the impairment parameter $\zeta$

$$\zeta \simeq \frac{Q_0^{(tum)}}{Q_0^{(nor)}} = \frac{Z_{tot}^{(nor)}}{Z_{tot}^{(tum)}}. \tag{2.6}$$

In figure 2b, we show an example of two histograms for $Z_{tot}^{(nor)}$ and $Z_{tot}^{(tum)}$ calculated for two stochastic networks of similar size and equal pressure difference $\Delta P$. It is apparent that $\zeta < 1$ in most of the resulting outcomes.

## 2.4. Heat loss

In order to quantify heat loss to the environment, we assume that its variation is a function of blood temperature, which in the present model will act as a surrogate for the body's internal temperature. We consider two ways of losing energy to the environment: one by evaporative cooling (via sweating) and the second by convection on the skin [38]. Variation of sweat rate is taken to be as in [39]

$$Q_e = h_e A(T_b - T_{beq}), \tag{2.7}$$

where we have neglected the contribution of temperature variation in the skin.

Finally, we model the variation of heat loss by convection on the skin. When the body's internal temperature increases, it reacts by diverting blood flow to the skin, which leads to vessel dilation through hypothalamic signalling. We account for this effect by varying the heat transfer coefficient $h$ according to the blood temperature $T_b$, which can be thought of as a surrogate for core temperature. By resorting to the fit for the relative blood flow in the skin, given in [38], which used data from [40], we assume that the heat transfer coefficient varies depending on the blood flow temperature, i.e.

$$h = h_0 \left[ 1 + \kappa(T_b - T_{beq}) - \nu(T_b - T_{beq})^2 \right], \tag{2.8}$$

where $\kappa$ and $\nu$ are constants given in table 1.

**Table 1.** Variables and parameters of the mathematical model.

| description | parameter | value | units | references |
|---|---|---|---|---|
| density of the tumour and surrounding tissue | $\rho, \rho_s$ | 1050 | kg m$^{-3}$ | [41] |
| heat capacity of the tumour and surrounding tissue | $c, c_s$ | 3.86 | MJ °C$^{-1}$ m$^{-3}$ | [42] |
| heat capacity of blood | $c_b$ | 3.82 | MJ °C$^{-1}$ m$^{-3}$ | [42] |
| average heat capacity of the entire body | $c_T$ | 4.12 | MJ °C$^{-1}$ m$^{-3}$ | [42] |
| total mass of the body | $m_T$ | 70 | kg | [42] |
| surface area of the body | $A$ | 1.91 | m$^2$ | [43] |
| volume of the tumour | $V$ | 80 | cm$^3$ | — |
| volume of the surrounding tissue (brain) | $V_s$ | 1200 | cm$^3$ | [44] |
| metabolic heat of the body in the rest state | MET | 85 | W | [42,45] |
| metabolic heat production of the surrounding tissue and tumour tissue | $Q_s, Q$ | 11 | W kg$^{-1}$ | [46,47] |
| effective power within the tissue | $P, P_s$ | 40 | W kg$^{-1}$ | [45] |
| escalation parameter for sweat rate | $h_e$ | 80.25 | W m$^{-2}$ °C$^{-1}$ | [39] |
| adjustment parameter of heat loss by convection with changing temperature | $\kappa, \nu$ | 8.25, 3.8 | °C$^{-1}$, °C$^{-2}$ | [38,40] |
| homeostatic equilibrium temperature of human brain | $T_{seq}$ | 36.9 | °C | [48] |
| temperature of the environment | $T_E$ | 22 | °C | — |
| baseline value for blood flow | $w_0$ | 49.2 | ml min$^{-1}$ per 100 ml of tissue | [49] |
| proportionality of the change of blood flow with temperature | $\chi$ | 0.1 | °C$^{-1}$ | [33] |
| proportionality of the change of blood flow with temperature in tumour | $\gamma$ | [0–0.1] | °C$^{-1}$ | — |
| parameter for blood flow impairment in tumour tissue | $\zeta$ | [0.25–1.2] | — | — |

## 2.5. Implementation and numerical solution

Our model equations were implemented using the scientific software environment Matlab (R2018b, MathWorks) and run on a 16 GB memory 3.2 GHz iMac machine. The code is available for public access on the websites specified in the Data accessibility section. We developed the wrapper HTforBrainTumour to frame all the different conditions presented in this paper, keep values of the parameters, and enclose the data flow. The parameters are grouped in different data structs related to different categories (geometrical, heat transfer, applied power, etc.) which in the default setting contain the values from table 1. These data structs are passed from the higher-level function HTforBrainTumour to EqSolver whenever the solution of equations (2.1) and (2.2) is needed. EqSolver performs the calculations and returns the results back to the main function, which in turn calls specific graphical routines. In order to provide a more realistic scenario, stochastic fluctuations of the blood flow that would happen in a real tumour [50] were added. These fluctuations are generated through a multiplicative factor of the blood flow that consists of a purely harmonic part with a 30 min period, and uniformly distributed random fluctuations centred around zero that vary every 1.5 min. The amplitude of the latter depends on the parameter $\zeta$, being larger for increasingly impaired vasculatures (i.e. for lower $\zeta$), and the evolution is taken to be linear between consecutive changes. To solve the model equations describing the evolution of temperatures and blood flow, EqSolver resorts to the ode23tb built-in Matlab function, which deals with stiff differential equations using a combination of the trapezoidal rule and a backward differentiation scheme. The function is based on the algorithm TR-BDF2 (trapezoidal rule-second order backward differentiation formula) developed by Randolph E. Bank and co-workers for the simulation of circuits in semiconductor devices [51,52]. The actual expression of the equations that EqSolver uses is coded in Matlab language in the function ThermalModel.

# 3. Results

## 3.1. Parametrization

Solving the model equation (2.1), allows us to evaluate the temperature changes and blood flow during and after HT. We consider the values of tissue density and heat capacity of the tumour tissue and its surrounding tissue to be equal, with values of 1050 kg m$^{-3}$ for densities $\rho_s$ and $\rho$ [41], and 3.86 MJ °C$^{-1}$ m$^{-3}$ for the heat capacities $c_s$ and $c$ [42]. Additionally, we use a value of 3.82 MJ °C$^{-1}$ m$^{-3}$ for the heat capacity of the blood $c_b$, and 4.12 MJ °C$^{-1}$ m$^{-3}$ as an average for the heat capacity of the entire body $c_T$ [42]. The average human body mass is taken to be $m_T = 70$ kg [42] and its surface area $A = 1.91$ m$^2$ [43]. We consider a tumour having a volume $V$ of 80 cm$^3$ located in a brain of 1280 cm$^3$ [44]. The surrounding tissue is taken to be the whole brain since the heating device is considered non-specific, as it happens with some capacitive heating devices, which are the most widespread methods of applying loco-regional hyperthermia [53] due to its ease of use and simplicity [54]. Therefore, the heated surrounding tissue $V_s$ is 1200 cm$^3$. The metabolic heat produced by the body at rest (e.g. during hyperthermia administration) is approximately 85 W [42,45], which is the value taken for MET, while the brain has a metabolic heat generation of 11 W kg$^{-1}$ ($Q_s$), corresponding to 21 mol ATP g$^{-1}$ min$^{-1}$ [46] and 30.5 kJ (mol ATP)$^{-1}$ [47]. For simplicity, the amount of heat ($Q$) produced in the tumour is assumed to be equal to that of the healthy brain ($Q_s$).

The proportionality $\chi$ is taken as 0.1 °C$^{-1}$ following [33]. For the blood flow we use a baseline value of 49.2 ml min$^{-1}$ per 100 ml of tissue [49]. As mentioned above, we use the parameter $\gamma$ for the proportionality between blood flow and temperature in tumours, which is expected to be less sensitive to temperature changes than $\chi$. Furthermore, the baseline value for the blood flow in the tumour will be different from that for the healthy tissue [35]. This is accounted for by parameter $\zeta$ which weighs the baseline perfusion value ($w_0$) with the one under pathological conditions. The parameter $\zeta$ is expected to be less than 1, which corresponds to a tumour having an impaired blood flow with respect to the healthy tissue, although we do not exclude the possibility that, in some cases, it may temporarily be greater than 1, for tumours characterized by an elevated blood flow [55].

Typical power values applied to tissue in HTs range between 20 and 40 W kg$^{-1}$ [45]. Since the brain is a very well-perfused organ, the elevation of temperature will be smaller than in other organs and therefore we choose 40 W kg$^{-1}$ for $P$, but will also explore the values of 25 and 50 W kg$^{-1}$. Henceforth, we consider a treatment with no focusing of power on the tumour, where the surrounding brain will receive the same power as the tumour, setting $P_s$ equal to $P$. Regarding the heat transfer to the environment through sweating and convection, which is modelled by equation (2.7), the sweating rate $h_e$ is taken to be 80.25 W m$^{-2}$ °C$^{-1}$ [39].

We performed a sensitivity analysis to evaluate the relative influence that variations of each parameter around its prescribed nominal value (as referred to in table 1) has on the output of our model. We used an iterative approximation based on adaptive increment directional derivatives of the output [56] as programmed by [57]. The system was analysed using the nominal values of every parameter, and the variation rate of the output temperatures of the model, due to perturbations of the parameters, was assessed. The relative percentage error in the output $\delta T$, due to the input uncertainty in each parameter $p_i$, denoted by $\Delta p_i$, was calculated as

$$\delta T = \frac{1}{T}\frac{\partial T}{\partial p_i}\Delta p_i = \frac{1}{T}\frac{\partial T}{\partial p_i}r|p_i|. \tag{3.1}$$

To make a meaningful inter-parameter comparison among all the $\Delta p_i$, we set an equal relative value $r = 10\%$ for all of them. The resulting influence of every parameter on each of the three output temperatures of our model are shown in table 2. It can be readily seen that the uncertainties of most of the parameters have a limited influence and that the most relevant ones are $T_{\text{seq}}$ that acts as a reference level, the baseline blood flow $w_0$ that is transporting heat, and the power level introduced in the system $P$. Note that the three resulting $\delta T$s, which are expressed as relative errors, did not exceed $r = 10\%$. This implies that the model is well defined and does not possess critical parameters in the explored parameter space.

In addition to that, we performed the same kind of analysis for the parameters defining the vascular state of the tumour $\gamma$ and $\zeta$. Since the range of variation of those is much broader (due to the limited information), we analysed all their range of variation. The results are shown in figure 3. The uncertainties in both of them have very little influence on the healthy tissue and the incoming blood

**Table 2.** Sensitivity analysis of the model parameters to determine their impact on the three temperatures (tumour, healthy tissue and blood compartments). The figures are relative percentage errors in each output for a 10% uncertainty in the considered parameter.

| parameter | $\delta T$ [%] | $\delta T_b$ [%] | $\delta T_s$ [%] |
|---|---|---|---|
| $\rho$ | $-1.3 \times 10^{-3}$ | $-1.9 \times 10^{-5}$ | $-2.2 \times 10^{-5}$ |
| $\rho_s$ | $-7.9 \times 10^{-5}$ | $-3.8 \times 10^{-5}$ | $-8.9 \times 10^{-5}$ |
| $C$ | $-1.3 \times 10^{-3}$ | $-1.9 \times 10^{-5}$ | $-2.2 \times 10^{-5}$ |
| $C_s$ | $-9.0 \times 10^{-5}$ | $-4.5 \times 10^{-5}$ | $-9.7 \times 10^{-5}$ |
| $C_b$ | $-6.5 \times 10^{-5}$ | $0.10$ | $-0.27$ |
| $C_T$ | $-5.8 \times 10^{-4}$ | $-3.7 \times 10^{-4}$ | $-4.2 \times 10^{-4}$ |
| $m_T$ | $-5.9 \times 10^{-4}$ | $-3.0 \times 10^{-4}$ | $-4.3 \times 10^{-4}$ |
| $A$ | $-2.4 \times 10^{-3}$ | $-3.1 \times 10^{-3}$ | $-2.6 \times 10^{-3}$ |
| $V$ | $8.4 \times 10^{-4}$ | $1.1 \times 10^{-3}$ | $9.2 \times 10^{-4}$ |
| $V_s$ | $0.01$ | $0.02$ | $0.01$ |
| MET | $-0.01$ | $-0.01$ | $-0.01$ |
| $Q$ | $0.19$ | $-1.7 \times 10^{-6}$ | $-2.1 \times 10^{-6}$ |
| $Q_s$ | $-0.09$ | $-0.1$ | $-0.01$ |
| $P$ | $0.56$ | $1.1 \times 10^{-3}$ | $9.3 \times 10^{-4}$ |
| $P_s$ | $0.01$ | $0.02$ | $0.29$ |
| $h_e$ | $-2.4 \times 10^{-3}$ | $-3.1 \times 10^{-3}$ | $-2.6 \times 10^{-3}$ |
| $\kappa$ | $-0.1$ | $-0.1$ | $-0.1$ |
| $\nu$ | $3.1 \times 10^{-4}$ | $4.2 \times 10^{-4}$ | $3.5 \times 10^{-4}$ |
| $T_{seq}$ | $9.17$ | $10.06$ | $9.65$ |
| $T_E$ | $-2.1 \times 10^{-4}$ | $-2.7 \times 10^{-4}$ | $-2.3 \times 10^{-4}$ |
| $w_0$ | $-0.65$ | $0.1$ | $-0.27$ |
| $\chi$ | $1.2 \times 10^{-5}$ | $5.9 \times 10^{-6}$ | $-0.04$ |

temperatures, although it does have some impact on the temperature of the tumour. Variations around the chosen nominal values of $\zeta$ become specially important as it decreases, while variations of $\gamma$ are more relevant as this parameter increases.

## 3.2. Equilibrium point and initial conditions

When no external power is applied ($P = 0$), the temperatures vary—due to random fluctuations in blood flow—around the following equilibrium points of the system (2.1)

$$T_{\mathrm{beq}} = T_E + \frac{\mathrm{MET}}{h_0 A}, \tag{3.2a}$$

$$T_{\mathrm{eq}} = T_{\mathrm{beq}} + \frac{Q}{\zeta w_0 c_b} \tag{3.2b}$$

and

$$T_{\mathrm{seq}} = T_{\mathrm{beq}} + \frac{Q_s}{w_0 c_b}. \tag{3.2c}$$

Equations (3.2) represent the thermal homeostatic conditions to which the unaltered system tends, i.e. the steady-state temperatures when there are no external energy sources. Typical brain temperatures $T_{\mathrm{seq}}$ are around 36.9°C [48]. From the known parameter values and by using equation (3.2c), we can solve for the temperature $T_{\mathrm{beq}}$ that the incoming blood must have to meet the physiological equilibrium conditions of the model. The same goes for the initial temperature of the tumour $T_{\mathrm{eq}}$ which can be calculated from equation (3.2b). The temperature of the environment ($T_E$) is set to 22°C and the value of the heat exchange parameter $h_0$ is chosen to equilibrate the metabolic heat for a given external temperature.

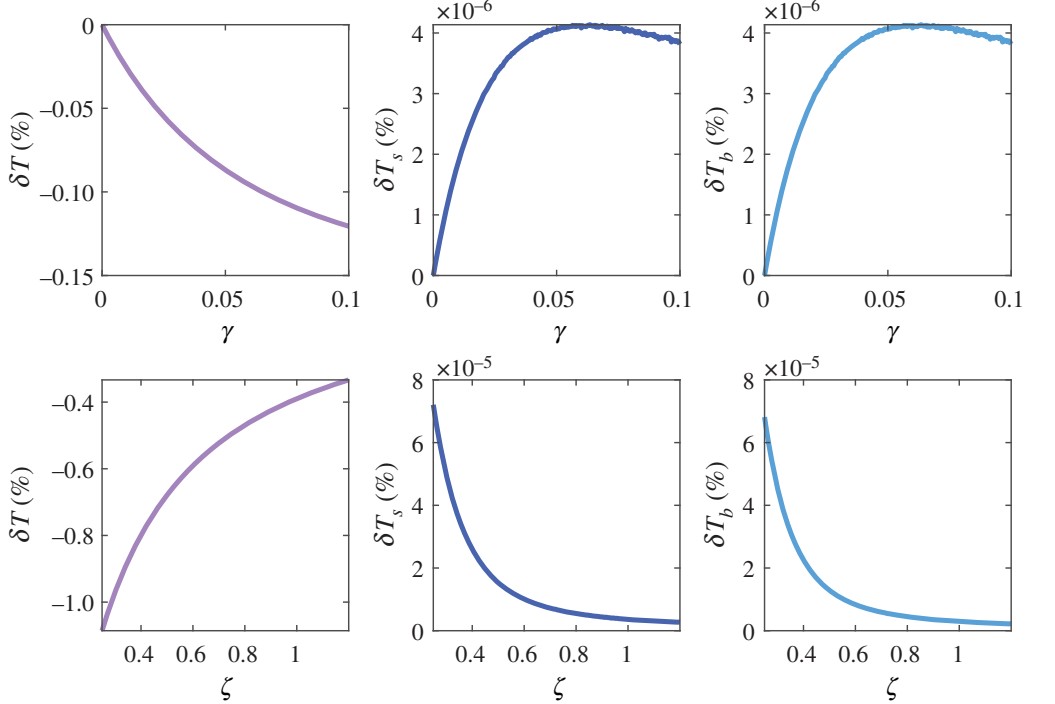

**Figure 3.** Results of the sensitivity analysis performed on the vascular parameters $\gamma$ and $\zeta$, related to the variation of tumour blood flow due to the elevation of temperature, and the impairment in the tumour blood flow with respect to that of the healthy tissue, respectively. The percentage errors $\delta T$, $\delta T_s$ and $\delta T_b$ in the corresponding output variables, due to the input uncertainty in each of the parameters, is depicted. The variation in any of these parameters shows a limited influence on the healthy tissue and blood temperatures (centre and right columns), while it is more prominent when considering the temperature of the tumour (left column). The uncertainty in the value of $\zeta$ becomes more important at lower values of the parameter, while for $\gamma$ the uncertainty gains relevance at high levels.

## 3.3. Computational simulation with a functional vasculature

First, we consider a tumour with a fairly intact vasculature that responds to temperature in a similar way to the healthy tissue and that has a similar level of baseline perfusion, but lower than the healthy tissue. According to this, we take $\zeta = 0.85 \pm 0.06$ and $\gamma = 0.08 \pm 0.008°C^{-1}$. The confidence intervals here represent the intervals within which the parameters lie in different simulations. We simulate a treatment lasting 55 min that starts at time $t = 5$ min and finishes at $t = 60$ min, with the first 10 min being allotted to progressively warming up the tissue (the power increases linearly from zero to the nominal value).

The results for the evolution of temperatures are shown in figure 4a, where the bands account for the envelopes of the temperatures obtained through multiple simulations using parameters within the indicated ranges, while the solid lines show the results with the nominal value of each parameter. The onset temperatures for the simulation are the pseudo-equilibrium values from equation (3.2) which are only perturbed by the variations of blood flow within the tumours. When the heating is started, there is an increase in temperature that follows the elevation of power. Shortly after the power has reached its full value, the tumour temperature attains a transient maximum. Neglecting the small fluctuations, a new equilibrium exists when the power is on that corresponds to the situation where the heat entering the system is the same as that lost by exchange with the environment through the skin. Therefore, there is a maximum temperature that can be reached, which is imposed by the conditions of the problem, and this maximum temperature is achieved early during heat administration. In this case, for a well-perfused tumour, the increase in temperature is 1.42°C and the observed plateau remains at values of mild hyperthermia, well below those usually reported of 42°C or higher. The temperature of the surrounding healthy tissue experiences a similar rise, but always lower than that of the tumour tissue due to its better ability to remove heat through its vasculature. A rise in the temperature of the entering blood is observed, albeit it is almost imperceptible. The maintenance of this temperature profile would be consistent with observed fact

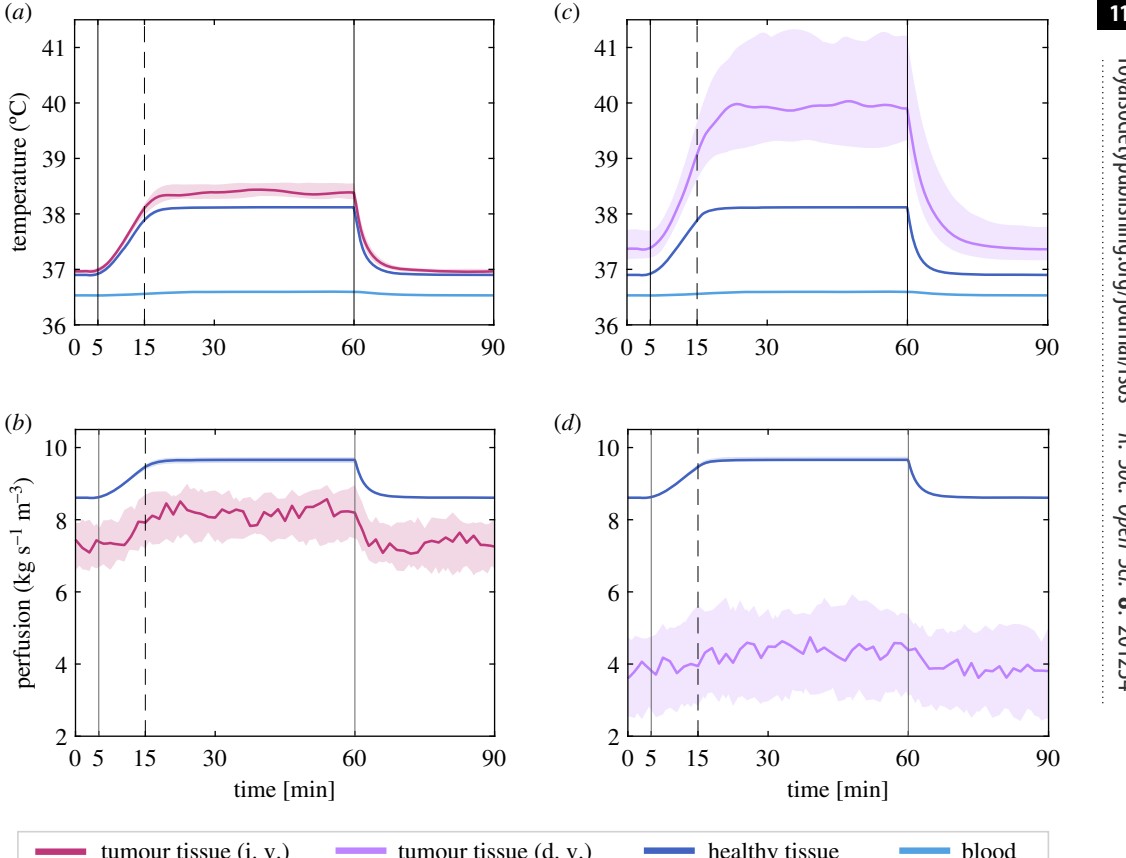

**Figure 4.** Left column: Evolution of temperatures (*a*) and perfusion (*b*) of an *in silico* brain tumour subjected to hyperthermia with an effective internal power density $P$ of 40 W kg$^{-1}$. The parameters correspond to a tumour with a fairly intact vasculature (i. v.) characterized by an impairment parameter $\zeta = 0.85 \pm 0.06$ and a response parameter $\gamma = 0.08 \pm 0.008°C^{-1}$. Treatment starts at minute 5 and power is increased until full power is reached at minute 15. The dark-pink curve corresponds to the tumour tissue while the dark-blue curve shows the values for the healthy surrounding tissue, which is also heated. The light-blue line represents the temperature of the blood entering into the tumour region and circulating around the body. The panels on the right display the same concepts—temperatures in (*c*), and blood flow in (*d*)—for a tumour with a dysfunctional vasculature (d. v., $\zeta = 0.44 \pm 0.13$, $\gamma = 0.06 \pm 0.006°C^{-1}$) whose evolution is shown by the purple line. Multiple runs are computed with different values of $\gamma$, $\zeta$ within their range of variation and the resulting envelope is represented by light colour bands. The result obtained with the nominal values of the parameters is plotted in solid line.

for the application of local hyperthermia, where the core body temperature of the patient remains at a fairly constant temperature.

The results for the evolution of perfusion in the tumour and in its surrounding tissue are shown in figure 4*b*. The increase in temperature produces a corresponding rise in blood flow, both in the tumour and in the healthy tissue. Since the vasculature of the healthy tissue is assumed to be more functional and responsive than that of the tumour, the values of perfusion are steadily higher. Leaving aside the fluctuations inherent in the tumour blood flow, the values in both tissues reach stability shortly after the onset of heating, matching the stabilization of temperature. After the cessation of the external source of heat, both the temperature and the blood flow return to their baseline figures at a high rate, doing so in less than 15 min.

We study an *in silico* tumour with similar characteristics defined in this case by the nominal values of the parameters and subjected to two more levels of maximum power, namely 25 and 50 W kg$^{-1}$, applied following the same administration mode. The results are shown in figure 5. The overall behaviour of temperature and perfusion exhibits the same progression, but it is apparent that the maximum temperature is highly affected by the power level, which is echoed by the levels of perfusion. However, even with the highest power the maximum achievable temperature is rather low by the standards of therapy, as is the time that it takes for the temperature and perfusion to return to basal levels.

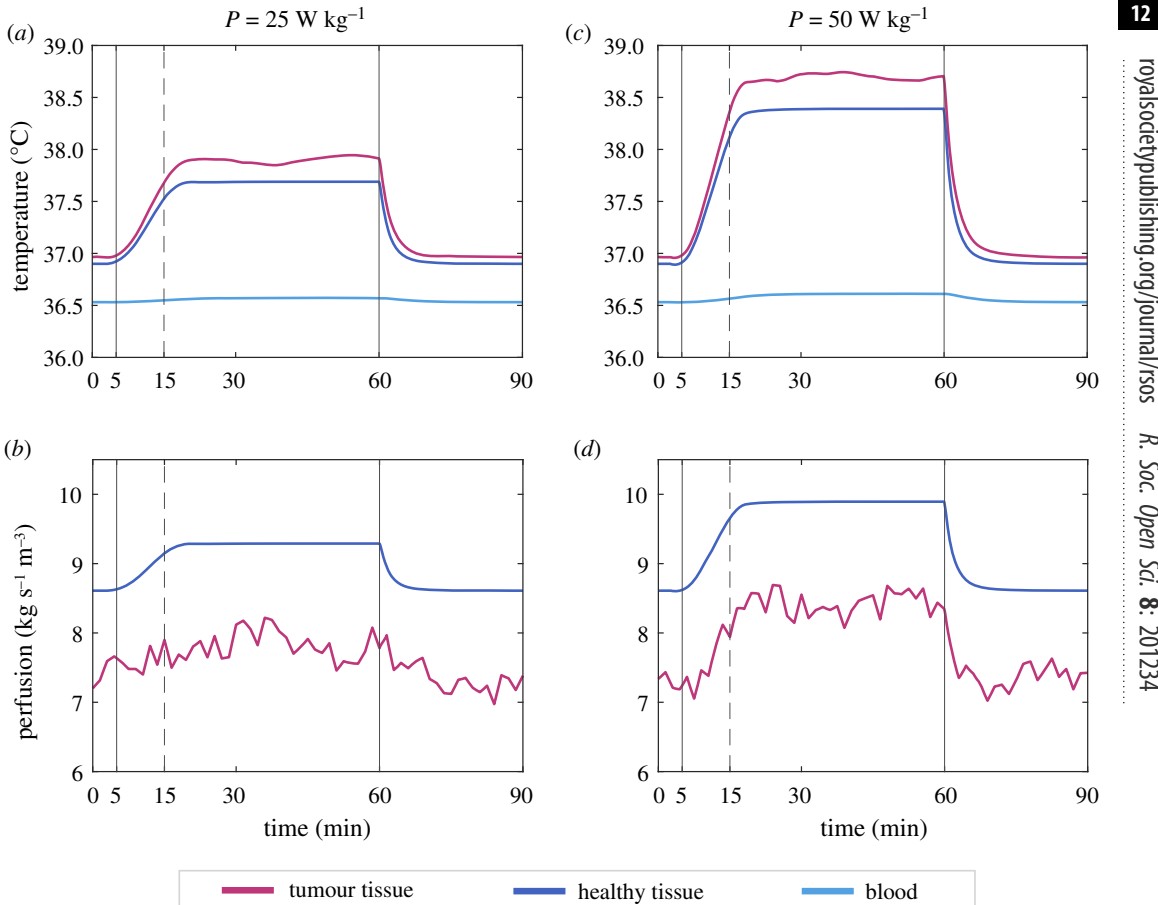

**Figure 5.** Different evolution of temperature and perfusion as a function of applied power. All the panels correspond to the case of a tumour with functional vascular status ($\zeta = 0.85$, $\gamma = 0.08°\text{C}^{-1}$). Panels (a) and (b) correspond to a heating with a maximum effective internal power density $P$ of 25 W kg$^{-1}$ administered from $t = 15$ min to $t = 60$ min, with a linear ramp from zero to the maximum power between $t = 5$ min and $t = 15$ min. The right-hand panels show the same concept for temperature (c), and perfusion (d) in a treatment with power density $P = 50$ W kg$^{-1}$.

## 3.4. Computational simulation with impaired vasculature

In the next scenario, we examine the heating process of a tumour with a dysfunctional perfusion due to a higher disruption of its vasculature. We use the parameter $\zeta = 0.44 \pm 0.13$ to quantify the impairment of tumour blood flow with respect to that of the normal tissue and the parameter $\gamma = 0.06 \pm 0.006°\text{C}^{-1}$ to describe the variation of blood flow for each degree change in temperature. The application of heat follows the same pattern as in the previous case (10 min warm-up with linearly increasing power and 45 more minutes at nominal power) and the results of temperature elevation and changes in perfusion for this case are shown in figure 4c,d, respectively. With impaired blood flow, the ability of the tissue to remove heat is lower and thus a higher temperature is achieved. It is apparent that the temperature of the tumour is much higher than that of the healthy tissue in this case. Nevertheless, the possibility of improving the perfusion in the tumour is not very high, since its response is lower than that of the healthy tissue. With a baseline perfusion of 3.79 ml kg s$^{-1}$ m$^{-3}$, the corresponding maximum blood flow obtained during the heating is 4.38 ml kg s$^{-1}$ m$^{-3}$. Moreover, the decrease after the cessation of the external power source is still very steep and both the blood flow and the temperature return to normal values in around 20 min (slightly longer than in the functional scenario).

To test the influence of the power density in this scenario, we ran the model with nominal values of the parameters ($\zeta = 0.44$, $\gamma = 0.06°\text{C}^{-1}$) for two values of power density, 25 and 50 W. The results are shown in figure 6 where it can be observed that power has a large influence on the temperatures that can be attained, but it does not significantly change the perfusion values. The attainable temperatures for this case are undoubtedly higher than for a tumour with a functional vasculature, but are still far

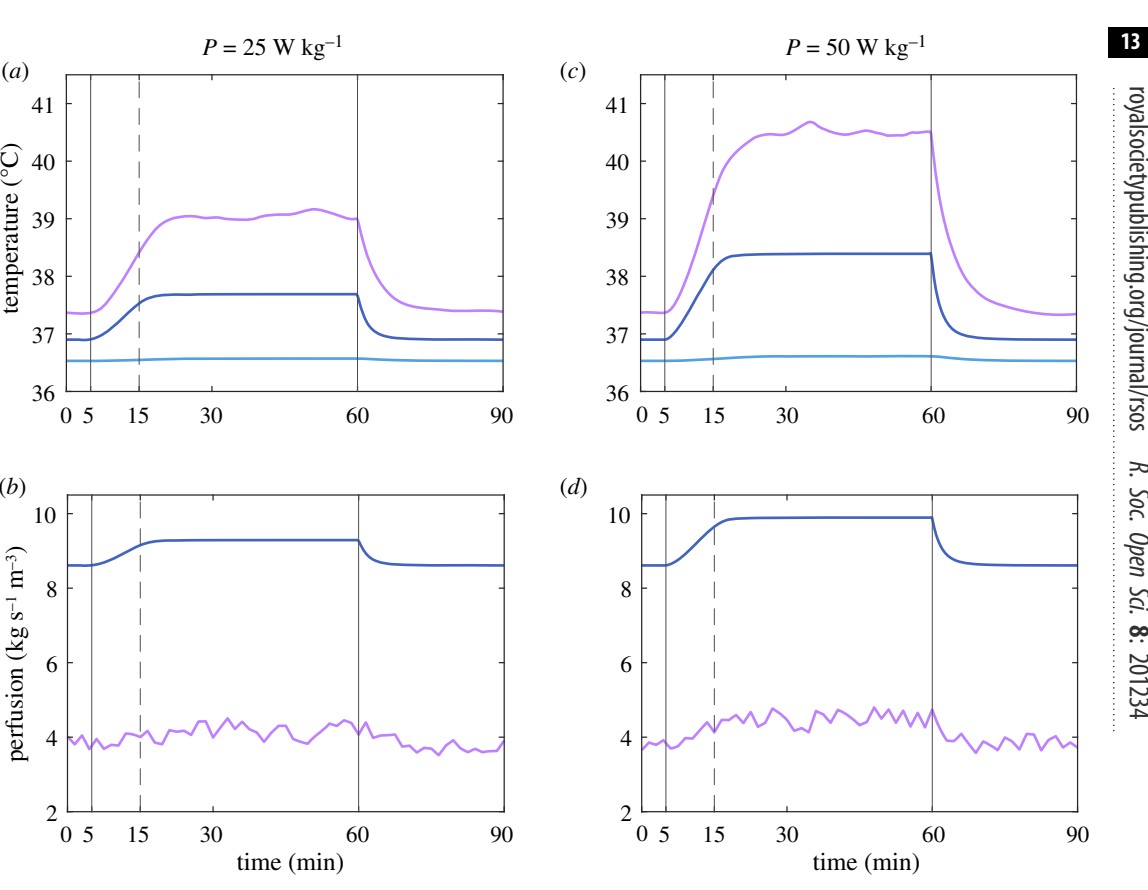

**Figure 6.** Difference in heating and blood perfusion for two different levels of applied powers on a tumour with impaired vasculature ($\zeta = 0.44$, $\gamma = 0.06°C^{-1}$). The left-hand column shows the temperature evolution in ($a$) and blood flow evolution in ($b$) for a tumour that undergoes hyperthermia treatment with an effective internal power $P$ of 25 W kg$^{-1}$ applied between $t = 15$ min and $t = 60$ min, and a linearly increasing power from $t = 5$ min until $t = 15$ min. ($c,d$) The same concept when the heating power is larger, $P = 50$ W kg$^{-1}$.

from the temperatures typically sought of around 43°C, which underlines how difficult it is to achieve therapeutically significant temperatures for such a well-perfused organ as the brain.

## 3.5. Map of variation of vascular parameters

Since the parameters that define the impairment of the flow ($\zeta$, $\gamma$) strongly affect the outcome of the treatment, we analyse their influence on the maximum temperature reached in the tumour during the treatment (figure 7). The parameter that defines the proportional variation of perfusion for each degree Celsius change in temperature ($\gamma$) is varied from 0, for a tissue that does not respond to temperature, to 0.1, a value observed in healthy brains [33]. The impairment in blood flow at the homeostatic temperature $\zeta$ ranges from 0.2, meaning that the tumour has a perfusion of about one fifth that of healthy tissue, to 1.2, for tumours that have an increased blood flow with respect to healthy tissue. The figure shows that the blood flow has a great impact on the temperatures that can be reached during the treatment. The parameter that shows the greatest influence is the initial impairment of the blood flow. Normal values similar to healthy tissue give maximum temperatures below 39°C, far from what is considered therapeutically relevant. To achieve temperatures greater than 41°C, the blood flow must be lower than about one-half of that of the surrounding healthy tissue, which corresponds to extremely impaired perfusion values; when $\zeta < 0.3$ this can lead to temperatures greater than 42°C. The effect of $\gamma$ on temperature is lower than that of $\zeta$, particularly when the latter takes normal values. However, lower values of $\gamma$ imply higher temperature values, since the blood flow is not able to respond to the rise in temperature as efficiently as healthy tissue.

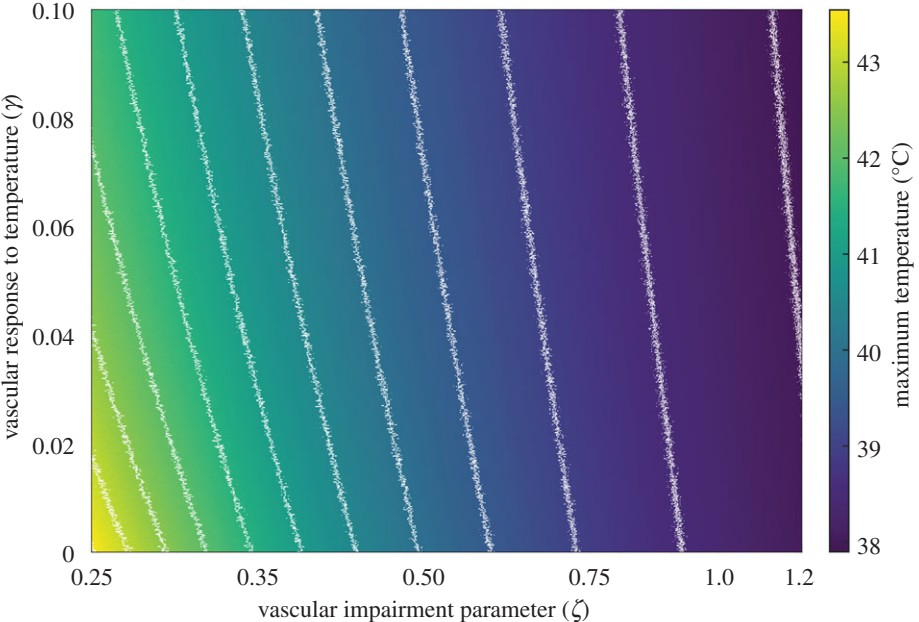

**Figure 7.** Map of temperatures in a tumour as a function of the vascular impairment parameters $\zeta$ and $\gamma$ entering in equation (2.2a). The parameter $\zeta$ accounts for the impairment in the baseline perfusion value ($w_0$), while $\gamma$ gives the proportional variation of the blood flow with each degree Celsius change in temperature, with the value ($\chi$) for the healthy tissue estimated to be $0.1°C^{-1}$. These parameters greatly influence the maximum temperatures reached.

## 3.6. Variation of basal perfusion

The previous results suggest that the typical values of blood flow in the brain make it very difficult to attain therapeutically relevant temperatures unless the vasculature of the tumour treated is highly impaired. Another immediate implication is that the time window during which the tumour region remains well above normal temperatures after local hyperthermia is very small (of the order of minutes). To analyse the response of the system to other values of the basal perfusion $w_0$, we ran simulations for a broad range of values of this parameter. Simulations were carried out using four different values of the vascular impairment parameter and for each pair of values ($w_0$, $\zeta$) we performed 14 different trials to account for the variability of the random effects of blood flow considered in our model. In these simulations, we quantified the maximum temperature reached over the heating time and the standard deviation (figure 8a). Temperatures higher than the usual target of 43°C were observed when both the basal perfusion and the vascular impairment parameter were lower. Additionally, in those simulations, we determined the time during which the tumour region remained hot after the cessation of the treatment. Results in figure 8b indicate that the time required to return to normal temperatures is very short whenever the values of perfusion are high, as already anticipated. As the basal perfusion of the organ decreases, the heated tumour remains at higher temperatures for longer times. However, none of the scenarios explored with our modelling approach support that the times of return to normal temperature might be higher than 1 h, thus setting an upper bound on the therapeutic time window.

## 4. Discussion

In this study, we put forward a mathematical model based on energy conservation laws to analyse the evolution of temperature and blood perfusion during regional hyperthermia of brain tumours (e.g. gliomas), motivated by the potential synergy with radiotherapy when treating this type of tumour, which is characterized by a low response and development of resistance [58]. Our model predicts a quasi-homeostatic equilibrium condition for the temperature of the brain and the blood vasculature feeding it, as given by equations (3.2a) and (3.2c), where the temperature of the brain is higher than the incoming blood by a number of 0.37°C, consistent with other models [32] and confirmed by experimental results [59]. In [31], the authors rationalize the shielding effect of blood flow in the brain

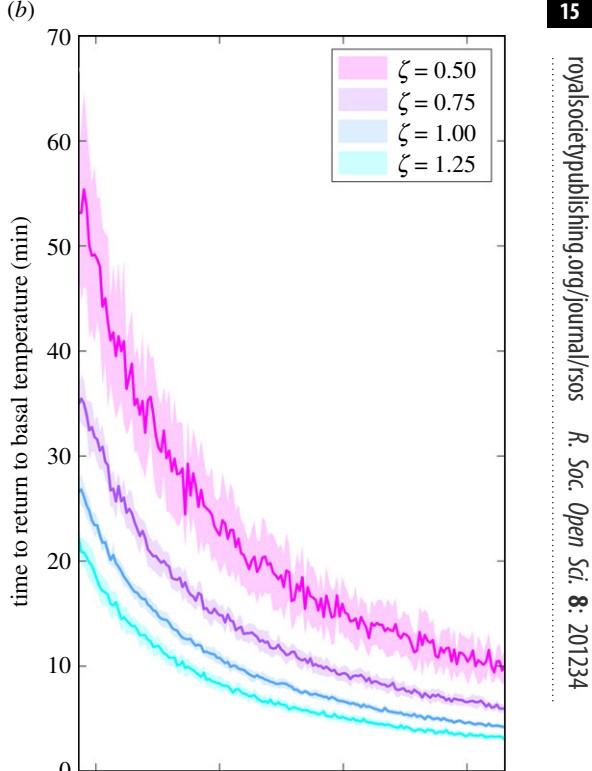

**Figure 8.** Maximum achievable temperature (*a*) and time that the tumour takes to return to normal temperature after the cessation of heating (*b*) as a function of basal perfusion $w_0$ for different values of the impairment parameter $\zeta$. Solid lines indicate the mean of 14 simulations for each ($w_0$, $\zeta$) pair and bands represent the standard deviation of each. Both maximum temperature and time to return to basal temperature are greatly improved when basal perfusion decreases.

which protects it from temperature variations induced by external cooling or heating, an effect that has been previously reported [60]. This effect, therefore, supports one of the assumptions of our model in which we attribute the variation of temperature in the heated tissue only to blood perfusion, and shift the dissipation of energy by conduction to the rest of the body. However, more detailed analyses of the heat losses might be needed to apply our model to different organs.

The elevation of temperature caused by HT is assumed to produce a subsequent rise in blood flow. Moriyama evaluated the CBF response to a rise in temperature in monkeys, finding a linear relation with the CBF increasing 10% for each degree of temperature rise [33]. Furthermore, in [34], the authors studied the decrease of CBF in dogs when exposed to cold temperatures and found a similar linear behaviour in which there was an approximate decrease of 7% in CBF for each degree of decrease, at least at temperatures not far from the homeostatic equilibrium. Taking these previous studies as a starting point, we have modelled the dependence of blood flow as a linear relation with temperature and assumed that the behaviour of tumour tissue would be similar but less reactive. While some authors have also used linear relations [61,62], others have resorted to more sophisticated functional forms connecting blood flow and temperature, such as in [63], although a linear relationship was finally employed. Alternatively, some authors have used other more complex empirical formulae [64,65] which ultimately derive from the experimental information contained in [66]. Nevertheless, what those models try to reproduce is mostly an extrapolation of the observed effects, since the actual data covers only a few points and does not seem to justify fits beyond those based on linear regression.

When hyperthermia applied to cancer started receiving attention at the end of the 1970s, the target temperatures were very high, usually higher than 43°C, with the objective of killing cells directly by the effect of heat [7]. Later, the interest turned to milder temperatures, below 41°C [5], which were easier to achieve and had the purpose of improving the synergy with chemo and radiotherapy [67]. The results of our model show that, due to the high blood flow that feeds the brain, it will be very difficult to achieve temperatures in the therapeutic range for brain tumours at the radiofrequency standard powers,

unless the vasculature of the tumour is highly compromised. A good knowledge of the pre-treatment blood flow is also essential. Our mathematical model keeps track of the temperature of the blood cooling the tissue, which is also circulating all over the body and losing heat to the environment. After solving the equations, it is observed that its temperature remains almost unaffected by the treatment and thus it could have been considered constant with no great difference in the results. As to the heated tissues, there is a pseudo-equilibrium around which the temperature oscillates, and that temperature is reached soon after starting the treatment. The pseudo-equilibrium appears due to the thermoregulation of the body, which is able to lose heat to the environment through the skin to compensate the power applied. This has direct relevance to the clinical setting, since it implies that, regarding the functionality of the vasculature, the temperature is bounded and there is control over the heating of the tissue.

The overall duration at which the vasculature is maintained at an elevated temperature may be important. In [68], the authors measured the evolution of CBF in adult male Wistar rats subjected to 8 MHz radio-frequency-induced localized cerebral hyperthermia. The results show a rise in CBF when the temperature is increased to temperatures lower than 43°C. After treatment ceases, CBF remains raised for some minutes, even though the temperature seems to return to baseline figures much faster. Also, in [69], the authors show that CBF might follow dynamics other than temperature when heating the cortex of eight-week-old male Sprague–Dawley rats. Therefore, we might encounter inertial effects not taken into account in the present form of our mathematical model. These mechanisms would be easy to add, provided that they are confirmed in humans and the correct parameters can be estimated. It must be taken into account that the sustained effect in CBF after hyperthermia might be only a reflection of increased heart rate, which stresses the importance of measuring heart rate in the patients receiving the treatment. Therefore, more experimental research on human patients is needed to confirm or reject the functional perfusion-temperature dependency that has been chosen here according to the current literature.

Another heating-mediated effect is the reduction in blood flow after exposure to heat [66,70,71]. This occurs when the temperatures reached are high enough or when the exposure time is long, pointing either to a destruction of the blood vessels caused by heat shock or to a vasoconstriction effect at high temperatures. Vujaskovic *et al.* suggested that a decrease in oxygenation was seen after hyperthermia in murine models at lower temperatures than in spontaneous tumours [72]. Again, this indicates a likely greater brittleness of rodent vessels (specially in xenograft models), which would attribute the decrease in oxygenation to the destruction of the vessels. The implications of this destruction would be enormous, since a reduction in blood flow implies a greater increase in temperature which feedbacks the process. Nonetheless, interest over the past few years has been directed at mild temperature hyperthermia rather than high temperatures (e.g. thermal ablation). Thus it seems more difficult that the potential damage to blood vessels is a main factor in the overall outcome of the treatment. Due to the low temperatures obtained with regular values of the parameters, we have not considered this possible destruction in our model. If it were to be included, a vasculature injury function of temperature and time should be derived based on experimental results that, for the moment, are not sufficiently systematized. This damage would act as a nonlinearity on the temperature, tending to increase its value when it is already sufficiently high.

To the best of our knowledge, no previous studies have clearly addressed the effects of heat on blood flow in humans. Due to the current availability of imaging techniques for non-invasively measuring blood flow [73], it would be advisable to test the differential response in perfusion after HT both in the tumour and in the healthy tissue. In [74], the authors presented one single case evaluating the perfusion in a head and neck cancer patient for whom there was a clear increase in blood flow immediately after heating the thorax using water-filtered IR-A radiation, and also five days after the treatment. This is a remarkable result that, if reproduced in other patients and further studied, would help to clarify many aspects of hyperthermia. Furthermore, it has been possible for many years to make use of magnetic resonance imaging techniques to evaluate the evolution of temperatures in tissues subjected to external heating non-invasively [75]. Despite the problems that may arise due to the obscuring influence of a changing perfusion [76,77], the evolution obtained from imaging, in combination with mathematical models that set the relations between the variables, may be very useful to clarify the open questions related to the operating mode of the treatment. In the same way, the utilization of two-point blood perfusion images [78], although not challenge-free [79], could elucidate current questions related to the evolution of blood flow under thermal treatment that have been addressed in this paper.

Traditionally, HT in oncology has been deemed to promote the increase in blood flow, which would lead to better oxygenation in the tissue [6], thereby improving the outcome of radiotherapy, which, as is well known, has improved efficiency in the presence of oxygen, in opposition to hypoxic microenvironments. Hyperthermia is said to be a very good way to overcome hypoxia, improving the

outcome of radiotherapy [2]. However, the actual role of improved perfusion is controversial [3,79]. Our computational model takes into account only mechanistic effects of heat on perfusion where a variation in temperature has immediate and direct effect on the modification of blood flow. Assuming this hypothesis, the results indicate that the decrease in temperature is very fast after the cessation of the external power, which would be accompanied by a parallel drop in blood flow. Usually, the application of radiotherapy after hyperthermia in the clinic is not immediate and takes place around an hour later [80]. There is currently controversy on whether a fast application of radiotherapy after hyperthermia does have an effect on the outcome [81,82]. These late works show the current lack of knowledge on which is the best scheduling for joint radio-hyperthermia, even though the work by Overgaard [83,84] has been very often cited as an answer to this regard. According to our results, considering the aforementioned clinical time gaps and the fast drop in temperature, it seems unlikely that improved oxygenation caused by an elevated perfusion is behind the benefits of HT, and other effects might have much higher relevance. A validation of the evolution of perfusion in humans patients who have undergone HT would very much clarify some of these current issues.

Our approach spatially averages the contribution in each of the compartments. The heterogeneous microenvironment typical of tumours ensures variability in the vascular conditions at different points that will lead to local differences in temperature and possibly overheated areas. At points where the vasculature is highly impaired or very leaky the flow might at times go to zero, creating hot-spots. These zones, however, would not benefit from an immediate improvement in oxygenation caused by an elevated temperature as their blood vessels cannot provide the required flow. Other local effects, like the induction of cell death followed by subsequent reoxygenation, or the turn to a glycolitic metabolism, might arise, but in these cases the time scales needed to achieve observable gains from a subsequent administration of radiotherapy would be much longer.

In conclusion, we have developed a mathematical model for the temperature and perfusion evolution of a brain tumour subjected to external heating. Our model emphasizes the importance of the status of tumour vasculature in the response to regional hyperthermia and points to the necessity of relating perfusion maps to patient outcome. Our results suggest that it may be difficult to significantly increase the temperature of such a well-perfused organ and show the existence of a quasi-steady-state maximum temperature that is reached fairly soon after the initiation of heat administration. Furthermore, the fast decay to baseline perfusion and temperature values after heating cessation apparently eliminates a raise in blood perfusion as a main factor for the favourable outcome of patients subjected to co-adjuvant hyperthermia with radiotherapy. Therefore, a mechanistic augmented perfusion due to heat does not appear to be the reason behind a better oxygenation, and the time scale of the latter would not correspond to the former. According to these results, other contributions seem more likely to be responsible and their time scales will be dominant for the combined treatment with radiotherapy.

Data accessibility. Data and relevant code for this research work are stored on the group's website: http://matematicas. uclm.es/molab/ThermalModelForHT.zip and have been archived within the Zenodo repository: https://doi.org/10. 5281/zenodo.4059158

Authors' contributions. J.J.B. and G.F.C. performed the analyses and drafted the manuscript. J.J.B. and M.C.N wrote the numerical code. G.F.C., V.M.P.-G. and M.C.N conceived and supervised the study and critically revised the manuscript. All authors gave final approval for publication.

Competing interests. We declare we have no competing interests.

Funding. This research has been supported by the James S. McDonnell Foundation Twenty-first Century Science Initiative in Mathematical and Complex Systems Approaches for Brain Cancer (Collaborative awards 220020560 and 220020450), Ministerio de Economía y Competitividad/FEDER, Spain (grant nos. MTM2015-71200-R), Junta de Comunidades de Castilla-La Mancha (grant nos. SBPLY/17/180501/000154 and SBPLY/19/180501/000211) and the Research grant no. PID2019-109652GB-I00 from the MICINN of the Spanish Government, which includes RDEF funds. J.J.B. is supported by funding from the University of Castilla-La Mancha (grant reference 2018-CPUCLM-7798).

Acknowledgements. The authors thank Juan Jiménez-Sánchez and Carmen Ortega-Sabater for their fruitful comments on the biology of cancer and the potential implications of thermal therapy.

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
