## [Reviewer comments · Royal Society Open Science]

Review History

RSOS-201234.R0 (Original submission)

Review form: Reviewer 1

Is the manuscript scientifically sound in its present form?

Yes

Are the interpretations and conclusions justified by the results?

Yes

Is the language acceptable?

Yes

Do you have any ethical concerns with this paper?

No

Have you any concerns about statistical analyses in this paper?

Yes

Recommendation?

Accept with minor revision (please list in comments)

Comments to the Author(s)

The authors present a well-written article on a well-designed mathematical model describing the blood flow and temperature for regional hyperthermia applications in the human brain. The suggested approach is well motivated and the mathematical model is a good balance of simplifying assumptions and detail modelled (some further simplifications may be possible as discussed by the authors). The model is applied to simulate the response of both tumour and healthy brain in terms of time-temperature and time-perfusion curves. The presented results are fairly expected and likely a consequence of the assumptions made in the model and the parameters used. This, in combination with the lack of experimental validation, despite the use of experimentally derived parameters, limits the clinical impact of the suggested model which was one of the study's aims. However, the model may provide a valid contribution towards simulations of combination treatments of hyperthermia and chemotherapy or radiation, that rely on an adequate description of a vascularized tumour in future studies and I hence recommend for publication given the inclusion for the points raised below. Most importantly, a sensitivity analysis and/or estimation of the contribution of uncertainty of the numerous model parameters to the model predictions should to be included, and the aims of the study should be given a slightly different angle for some aspects.

Detailed comments:

- Introduction/Discussion: One aim of the study was set out to be the identification of clinical limitations in current procedures and to give directions for possible improvements. It is unclear how this aim was achieved for some of the suggested conclusions: the conclusion of a fast cooling of the tumour and decrease in perfusion (hence optional shorter time between treatments) may not reflect the physiological scenario as pointed out by the authors in the discussion. Also, for combination treatments the inhibition of DNA repair may play an important role in addition to potential tumour reoxygenation (also discussed). Finally, the observation of an equilibrium temperature level is indeed interesting, but also relies on the assumption of a homogeneous treatment delivery and perfusion changes in a linear fashion. Despite the detailed discussion, neither oxygenation (only perfusion), nor an actual combination treatment, or a realistic treatment delivery were simulated here. Current challenges for Radiofrequency HT lie potentially rather in the estimation of the heat distribution, (homogeneous) treatment delivery and temperature monitoring which are not part of the model. The time between treatments may often be restricted due to patient logistics and set-up durations. The conclusion, that only tumours with a highly distorted vasculature could be subject to sufficient heating with the estimated power levels is based on the assumption of equal thermal and heating properties of the tumour and normal tissue, as well as the lack of any form of treatment localization and hence is based on a limited modelling of the actual treatment delivery. It is difficult, if not impossible to capture all complex biological processes in a single model and the presented approach is a good compromise indeed, but clinical conclusions should maybe be formulated less strongly. A valid aim of this study may also be to provide a modelling approach supported by the use of experimental parameters, which could be developed further for actual combination treatment effect simulations or as part of a more elaborate treatment delivery simulations.
- Section 2.1.: Why is no heat diffusion between the tumour and healthy brain considered?
- Section 2.2.: Vasodilation may be restricted to a low thermal dose range, for high thermal doses vasoconstriction occurs and blood flow is reduced (see e.g. <https://doi.org/10.1016/j.clon.2007.03.015>) – is a linear increase of blood flow with temperature

hence a suitable assumption since in some figures temperatures $>45^{\circ}\text{C}$ are reported? In the discussion the authors refer to heat-induced destruction of blood vessels only. Consider reporting thermal dose levels in addition to temperature alone.

- Include information on how the model was implemented (which coding platform and solvers).
- Define uncertainty bands in figure 3 – why is there no uncertainty for the normal tissue? Overall, parameter uncertainty would be essential to consider in the suggested simulations. Some figures include an uncertainty band due to variation of a single parameter (e.g. Figure 3), others (e.g. Figure 4) don't. All parameter uncertainty should be accounted for and a sensitivity analysis should be performed to inform on the importance of these parameter uncertainties. The relevant methods on how the uncertainty bands were calculated should be included.
- Comment on discussion: The authors refer to 'Standard power' and the hyperthermia technique assumed to deliver the treatment should be mentioned again (radio frequency). Some treatment techniques, such as high intensity focused ultrasound could locally easily achieve higher absorbed power levels – since the treatment delivery is not explicitly modelled, the approach could be extended to such applications in the low temperature hyperthermia regime, too, and power levels could then be higher.

Review form: Reviewer 2

Is the manuscript scientifically sound in its present form?

Yes

Are the interpretations and conclusions justified by the results?

Yes

Is the language acceptable?

Yes

Do you have any ethical concerns with this paper?

No

Have you any concerns about statistical analyses in this paper?

No

Recommendation?

Accept with minor revision (please list in comments)

Comments to the Author(s)

This is a very well-written paper which explains the problem well and builds a strong case for the hypotheses tested. The mathematical model developed supports the overall conclusion that a mechanistic augmented blood flow due to heat is almost certainly not the reason for a better oxygenation.

I have only a few very minor comments as detailed below:

P5. Line 40. "however, those contributions are neglected here as the temperature differences between the two tissues are typically sufficiently small." Please justify this by providing evidence.

P7. Line 47. "...an equal pressure difference ΔP can be established between some reference input arterial vessel and an output venous vessel." Please explain this assumption in more detail.

Code - whilst I appreciate that the code is made available open access, I would appreciate a dedicated section in the manuscript that briefly describes the procedure for numerically solving the equations (I.e. the methods used, verification etc.)

Decision letter (RSOS-201234.R0)

Dear Mr Bosque

On behalf of the Editors, we are pleased to inform you that your Manuscript RSOS-201234 "The interplay of blood flow and temperature in regional hyperthermia: A mathematical approach" has been accepted for publication in Royal Society Open Science subject to minor revision in accordance with the referees' reports. Please find the referees' comments along with any feedback from the Editors below my signature.

Please submit your revised manuscript and required files (see below) no later than 7 days from today's (ie 18-Sep-2020) date. Note: the ScholarOne system will 'lock' if submission of the revision is attempted 7 or more days after the deadline. If you do not think you will be able to meet this deadline please contact the editorial office immediately.

on behalf of Dr Oliver Jensen (Associate Editor) and Mark Chaplain (Subject Editor)

Associate Editor Comments to Author (Dr Oliver Jensen):

Associate Editor: 1

Comments to the Author:

Please revise your paper in light of the comments of both referees, paying careful attention to each point that they raise.

Reviewer comments to Author:

Reviewer: 1

Comments to the Author(s)

The authors present a well-written article on a well-designed mathematical model describing the blood flow and temperature for regional hyperthermia applications in the human brain. The suggested approach is well motivated and the mathematical model is a good balance of simplifying assumptions and detail modelled (some further simplifications may be possible as discussed by the authors). The model is applied to simulate the response of both tumour and healthy brain in terms of time-temperature and time-perfusion curves. The presented results are fairly expected and likely a consequence of the assumptions made in the model and the parameters used. This, in combination with the lack of experimental validation, despite the use of experimentally derived parameters, limits the clinical impact of the suggested model which was one of the study's aims. However, the model may provide a valid contribution towards simulations of combination treatments of hyperthermia and chemotherapy or radiation, that rely on an adequate description of a vascularized tumour in future studies and I hence recommend for publication given the inclusion for the points raised below. Most importantly, a sensitivity analysis and/or estimation of the contribution of uncertainty of the numerous model parameters to the model predictions should to be included, and the aims of the study should be given a slightly different angle for some aspects.

Detailed comments:

- Introduction/Discussion: One aim of the study was set out to be the identification of clinical limitations in current procedures and to give directions for possible improvements. It is unclear how this aim was achieved for some of the suggested conclusions: the conclusion of a fast cooling of the tumour and decrease in perfusion (hence optional shorter time between treatments) may not reflect the physiological scenario as pointed out by the authors in the discussion. Also, for combination treatments the inhibition of DNA repair may play an important role in addition to potential tumour reoxygenation (also discussed). Finally, the observation of an equilibrium temperature level is indeed interesting, but also relies on the assumption of a homogeneous treatment delivery and perfusion changes in a linear fashion. Despite the detailed discussion, neither oxygenation (only perfusion), nor an actual combination treatment, or a realistic treatment delivery were simulated here. Current challenges for Radiofrequency HT lie potentially rather in the estimation of the heat distribution, (homogeneous) treatment delivery and temperature monitoring which are not part of the model. The time between treatments may often be restricted due to patient logistics and set-up durations. The conclusion, that only tumours with a highly distorted vasculature could be subject to sufficient heating with the estimated power levels is based on the assumption of equal thermal and heating properties of the tumour and normal tissue, as well as the lack of any form of treatment localization and hence is based on a limited modelling of the actual treatment delivery. It is difficult, if not impossible to capture all complex biological processes in a single model and the presented approach is a good compromise indeed, but clinical conclusions should maybe be formulated less strongly. A valid aim of this study may also be to provide a modelling approach supported by the use of experimental

parameters, which could be developed further for actual combination treatment effect simulations or as part of a more elaborate treatment delivery simulations.

- Section 2.1.: Why is no heat diffusion between the tumour and healthy brain considered?

- Section 2.2.: Vasodilation may be restricted to a low thermal dose range, for high thermal doses vasoconstriction occurs and blood flow is reduced (see e.g. <https://doi.org/10.1016/j.clon.2007.03.015>) – is a linear increase of blood flow with temperature hence a suitable assumption since in some figures temperatures >45C are reported? In the discussion the authors refer to heat-induced destruction of blood vessels only. Consider reporting thermal dose levels in addition to temperature alone.

- Include information on how the model was implemented (which coding platform and solvers).

- Define uncertainty bands in figure 3 – why is there no uncertainty for the normal tissue? Overall, parameter uncertainty would be essential to consider in the suggested simulations. Some figures include an uncertainty band due to variation of a single parameter (e.g. Figure 3), others (e.g. Figure 4) don't. All parameter uncertainty should be accounted for and a sensitivity analysis should be performed to inform on the importance of these parameter uncertainties. The relevant methods on how the uncertainty bands were calculated should be included.

- Comment on discussion: The authors refer to 'Standard power' and the hyperthermia technique assumed to deliver the treatment should be mentioned again (radio frequency). Some treatment techniques, such as high intensity focused ultrasound could locally easily achieve higher absorbed power levels – since the treatment delivery is not explicitly modelled, the approach could be extended to such applications in the low temperature hyperthermia regime, too, and power levels could then be higher.

Reviewer: 2

Comments to the Author(s)

This is a very well-written paper which explains the problem well and builds a strong case for the hypotheses tested. The mathematical model developed supports the overall conclusion that a mechanistic augmented blood flow due to heat is almost certainly not the reason for a better oxygenation.

I have only a few very minor comments as detailed below:

P5. Line 40. "however, those contributions are neglected here as the temperature differences between the two tissues are typically sufficiently small." Please justify this by providing evidence.

P7. Line 47. "...an equal pressure difference ΔP can be established between some reference input arterial vessel and an output venous vessel." Please explain this assumption in more detail.

Code - whilst I appreciate that the code is made available open access, I would appreciate a dedicated section in the manuscript that briefly describes the procedure for numerically solving the equations (I.e. the methods used, verification etc.)

===PREPARING YOUR MANUSCRIPT===

===PREPARING YOUR REVISION IN SCHOLARONE===

- An editable file of each table (.doc, .docx, .xls, .xlsx, or .csv).
- An editable file of all figure and table captions.

- Any electronic supplementary material (ESM).
- If you are requesting a discretionary waiver for the article processing charge, the waiver form must be included at this step.
- If you are providing image files for potential cover images, please upload these at this step, and inform the editorial office you have done so. You must hold the copyright to any image provided.
- A copy of your point-by-point response to referees and Editors. This will expedite the preparation of your proof.

- Ensure that your data access statement meets the requirements at <https://royalsociety.org/journals/authors/author-guidelines/#data>. You should ensure that you cite the dataset in your reference list. If you have deposited data etc in the Dryad repository, please only include the 'For publication' link at this stage. You should remove the 'For review' link.
- If you are requesting an article processing charge waiver, you must select the relevant waiver option (if requesting a discretionary waiver, the form should have been uploaded at Step 3 'File upload' above).
- If you have uploaded ESM files, please ensure you follow the guidance at <https://royalsociety.org/journals/authors/author-guidelines/#supplementary-material> to include a suitable title and informative caption. An example of appropriate titling and captioning may be found at [https://figshare.com/articles/Table_S2_from_Is_there_a_trade-off_between_peak_performance_and_performance_breadth_across_temperatures_for_aerobic_sc](https://figshare.com/articles/Table_S2_from_Is_there_a_trade-off_between_peak_performance_and_performance_breadth_across_temperatures_for_aerobic_scope_in_teleost_fishes_/3843624) ope_in_teleost_fishes_/3843624.

Author's Response to Decision Letter for (RSOS-201234.R0)

See Appendix A.

RSOS-201234.R1 (Revision)

Review form: Reviewer 1

Is the manuscript scientifically sound in its present form?

Yes

Are the interpretations and conclusions justified by the results?

Yes

Is the language acceptable?

Yes

Do you have any ethical concerns with this paper?

No

Have you any concerns about statistical analyses in this paper?

No

Recommendation?

Accept with minor revision (please list in comments)

Comments to the Author(s)

The authors have included all the recommended changes and provide a very thorough analysis that is very interesting. The model and the conclusions drawn seem reasonable and relevant to the community. A few minor points could briefly be addressed:

Figure 1 and 2: The letters A to D are very large compared to the normal text.

Figure 1: in the caption the reference to the equation is missing (??).

p.8. Include the actual reference to the group's website.

p.8.: How were the stochastic fluctuations of blood flow included? (What probabilities)

p.8: What about the hyperparameters of EqSolver (assumingly default?)

Figure 4/5/6: Although given in the text, a legend in the figure would help to quickly assess which lines are which tissue. In Figure 5 indicate the power level in the figure, too. There is a line under the degree symbol on the y-axis.

p.18.1.20: Correct 'Or well' in blue text

Published code: A brief 'readme' file outlining the different functions, their role, and how they should be executed together would help potential users.

Decision letter (RSOS-201234.R1)

Dear Mr Bosque

On behalf of the Editors, we are pleased to inform you that your Manuscript RSOS-201234.R1 "The interplay of blood flow and temperature in regional hyperthermia: A mathematical approach" has been accepted for publication in Royal Society Open Science subject to minor revision in accordance with the referees' reports. Please find the referees' comments along with any feedback from the Editors below my signature.

We invite you to respond to the comments and revise your manuscript. Below the referees' and Editors' comments (where applicable) we provide additional requirements. Final acceptance of

your manuscript is dependent on these requirements being met. We provide guidance below to help you prepare your revision.

Please submit your revised manuscript and required files (see below) no later than 7 days from today's (ie 03-Nov-2020) date. Note: the ScholarOne system will 'lock' if submission of the revision is attempted 7 or more days after the deadline. If you do not think you will be able to meet this deadline please contact the editorial office immediately.

on behalf of Dr Oliver Jensen (Associate Editor) and Mark Chaplain (Subject Editor)
openscience@royalsociety.org

Associate Editor Comments to Author (Dr Oliver Jensen):

Associate Editor: 1

Comments to the Author:

Please address the remaining minor suggestions from the reviewer.

Reviewer comments to Author:

Reviewer: 1

Comments to the Author(s)

The authors have included all the recommended changes and provide a very thorough analysis that is very interesting. The model and the conclusions drawn seem reasonable and relevant to the community. A few minor points could briefly be addressed:

Figure 1 and 2: The letters A to D are very large compared to the normal text.

Figure 1: in the caption the reference to the equation is missing (??).

p.8. Include the actual reference to the group's website.

p.8.: How were the stochastic fluctuations of blood flow included? (What probabilities)

p.8: What about the hyperparameters of EqSolver (assumingly default?)

Figure 4/5/6: Although given in the text, a legend in the figure would help to quickly assess which lines are which tissue. In Figure 5 indicate the power level in the figure, too. There is a line under the degree symbol on the y-axis.

p.18.1.20: Correct 'Or well' in blue text

Published code: A brief 'readme' file outlining the different functions, their role, and how they should be executed together would help potential users.

===PREPARING YOUR MANUSCRIPT===

===PREPARING YOUR REVISION IN SCHOLARONE===

<https://royalsociety.org/journals/authors/author-guidelines/#supplementary-material> to include a suitable title and informative caption. An example of appropriate titling and captioning may be found at https://figshare.com/articles/Table_S2_from_Is_there_a_trade-off_between_peak_performance_and_performance_breadth_across_temperatures_for_aerobic_scops_in_teleost_fishes_/3843624.

Author's Response to Decision Letter for (RSOS-201234.R1)

See Appendix B.

Decision letter (RSOS-201234.R2)

Dear Mr Bosque,

It is a pleasure to accept your manuscript entitled "The interplay of blood flow and temperature in regional hyperthermia: A mathematical approach" in its current form for publication in Royal Society Open Science. The comments of the reviewer(s) who reviewed your manuscript are included at the foot of this letter.

on behalf of Dr Oliver Jensen (Associate Editor) and Mark Chaplain (Subject Editor)
openscience@royalsociety.org

Appendix A

Ciudad Real, Spain
25 September 2020

Author Reply of Manuscript **RSOS-201234** submitted to **R. S. Open Science**,
Title: *The interplay of blood flow and temperature in regional hyperthermia:
A mathematical approach*,
Authors: Jesús J. Bosque, Gabriel F. Calvo, Víctor M. Pérez-García and
María Cruz Navarro

Dear Dr Jensen

We have examined the reports from the reviewers and submit a revised manuscript which accounts for all the points raised by them. We would like to thank the reviewers for their time, helpful comments and suggestions. Additionally, we would like to thank the Editorial Office of Royal Society Open Science for considering our manuscript for publication.

In what follows we detail our responses to each of the reviewer's comments and suggestions and list the changes made to the manuscript. Changes to the manuscript have been highlighted in blue.

Yours sincerely,

Jesús J. Bosque

Department of Mathematics
MOLAB-Mathematical Oncology Laboratory
Universidad de Castilla-La Mancha
Ciudad Real, Spain

Reviewer 1

We would like to thank the reviewer for his/her careful reading of the manuscript and his/her very useful comments which certainly have helped us to improve our manuscript. We now proceed to reply to all the points raised and describe the changes made in the manuscript, which have been highlighted in blue colour in the main text.

Comment: [...] *Clinical conclusions should maybe be formulated less strongly.*

Reply: We have softened the statements related to the clinical conclusions both in the Abstract and at the end of the Discussion.

Comment: *Why is no heat diffusion between the tumour and healthy brain considered?*

Reply: In earlier versions of our model we did consider this process. However, in the light of the research performed by A.L. Sukstanskii and D.A. Yablonskiy (references [31] and [32] of the manuscript), together with previous experimental results (ref. [55]), confirmed that blood flow is the main regulatory mechanism of the temperature of the brain. The blood flow carries away thermal energy and generates a “shielding effect” that makes conductive variations important only to very short distances. Therefore, the diffusive term may be safely neglected. We made use of this fact to give rise to a simple ODE system that only depends on time. Moreover, removing this simplifying assumption would actually strengthen our findings, since the heat losses in the tumour would be higher, contributing to a lower temperature and lower return times after the treatment, thus reinforcing the conclusions of our paper. This point was already present in the first paragraph of our Discussion and in the third paragraph of Section 2.1. However, we have added some comments in the latter to clarify this issue.

Comment: *Vasodilation may be restricted to a low thermal dose range, for high thermal doses vasoconstriction occurs and blood flow is reduced (see e.g. <https://doi.org/10.1016/j.clon.2007.03.015>) is a linear increase of blood flow with temperature hence a suitable assumption since in some figures temperatures $> 45C$ are reported? In the discussion the authors refer to heat-induced destruction of blood vessels only. Consider reporting thermal dose levels in addition to temperature alone.*

Reply: The reviewer is right, the occurrence of high temperatures might induce further effects beyond the scope of our model. We do not intend to create a general method with applicability to a broad range of temperatures, but rather focus on the effects of mild temperature hyperthermia. Therefore, our assumptions are set and discussed taking that range into consideration. Consequently, and in order to avoid confusion in the interpretation, we have modified the previous Figure 6 (currently Figure 7) of the manuscript so that all the temperatures shown fall within the range of interest and applicability. Furthermore, the reviewer is right in pointing that this reduction could also be due to vasoconstriction, and consequently we have included that possibility in the corresponding paragraph of our Discussion.

Comment: *Include information on how the model was implemented (which coding platform and solvers).*

Reply: In the revised manuscript we have added a Section 2.5 aimed at providing the key aspects of the numerical method used to solve the equations and some technical details of the procedure.

Comment: *Define uncertainty bands in figure 3 why is there no uncertainty for the normal tissue? Overall, parameter uncertainty would be essential to consider in the suggested simulations. Some figures include an uncertainty band due to variation of a single parameter (e.g. Figure 3), others (e.g. Figure 4) dont. All parameter uncertainty should be accounted for and a sensitivity analysis should be performed to inform on the importance of these parameter uncertainties. The relevant methods on how the uncertainty bands were calculated should be included.*

Reply: To assess how much the variation/uncertainty of each parameter influences the output of the system, we have performed a sensitivity analysis that is now included in the revised manuscript, incorporating a table with the results as well as a new figure that is now Figure 3.

The way the results from the former Figure 3 (now Figure 4) were generated is as follows: We ran 25 simulations with varying values of parameters γ and ζ around the centred values indicated in the caption and within the confidence interval given. After performing these simulations, we represented the bands within which the output temperatures lie, i.e. the envelope of all the results, and plotted a solid line for the obtained result corresponding to the nominal values of the parameters. The bands have been represented for all the output temperatures shown in the graph. However, the width of the band is almost imperceptible for the surrounding healthy tissue and blood temperatures, due to the weak dependency of these variables on the parameters γ and ζ .

Figures 4 and 5 (now Figures 5 and 6) do not include uncertainty bands. The reason is that our aim was to elucidate how the system would respond when subjected to different levels of power following the same application scheme. Thus, we focused on the power rather than on other parameters. As a consequence, we performed one only simulation for each set of nominal values of the parameters. Although we did evaluate two separate cases of different γ and ζ , the focus was not set on those and we found more instructive to prescind of any uncertainty on them.

To meet the requirements outlined by the referee, we have incorporated a sensitivity analysis in our manuscript with the results collected in a new Table 2 and in a new Figure 3. We have also included information about the used methods in Section 3.3. of the Results and added complementary information in the caption of the current Figure 4.

Comment: *The authors refer to Standard power and the hyperthermia technique assumed to deliver the treatment should be mentioned again (radio frequency).*

Reply: We have changed the words 'standard powers' to 'radiofrequency standard powers', as the reviewer suggested.

Reviewer 2

We would like to thank the reviewer for his/her time and helpful comments. In what follows, we answer to the points raised and summarize the changes made to the revised manuscript, which have been highlighted in blue colour.

Comment: [...] *however, those contributions are neglected here as the temperature differences between the two tissues are typically sufficiently small. Please justify this by providing evidence.*

Reply: In earlier versions of our model we did consider this process. However, in the light of the research performed by A.L. Sukstanskii and D.A. Yablonskiy (references [31] and [32] of the manuscript), together with previous experimental results (ref. [55]), confirmed that blood flow is the main regulatory mechanism of the temperature of the brain. The blood flow carries away thermal energy and generates a “shielding effect” that makes conductive variations important only to very short distances. Therefore, the diffusive term may be safely neglected. We made use of this fact to give rise to a simple ODE system that only depends on time. Moreover, removing this simplifying assumption would actually strengthen our findings, since the heat losses in the tumour would be higher, contributing to a lower temperature and lower return times after the treatment, thus reinforcing the conclusions of our paper. This point was already present in the first paragraph of our Discussion and in the third paragraph of Section 2.1. However, we have added some comments in the latter to clarify this issue.

Comment: [...]...*an equal pressure difference ΔP can be established between some reference input arterial vessel and an output venous vessel. Please explain this assumption in more detail*

Reply: Here ΔP is a pressure difference that acts as a reference to compare two sufficiently large but different vascular networks working under identical pressure differences. Physiologically this matches the condition of a tissue that is being fed by an arterial input with a given pressure propelled by the heart and draining to a vein at a lower pressure imposed by the rest of the parallel circuit of the vascular network. Therefore, the performance of different vascular systems can be compared attending to their behaviour under the same reference pressure difference. A brief commentary has been added in that part of the revised manuscript to try to clarify that point.

Comment: *I would appreciate a dedicated section in the manuscript that briefly describes the procedure for numerically solving the equations (I.e. the methods used, verification etc.*

Reply: In the revised manuscript we have added a Section 2.5 aimed at providing the key aspects of the numerical method used to solve the equations and some technical details of the procedure.

This concludes our reply to the reviewers comments.

Sincerely,

The authors

Appendix B

Ciudad Real, Spain
11 November 2020

Author Reply of Manuscript **RSOS-201234**, **second revision**, submitted to
R. S. Open Science,

Title: *The interplay of blood flow and temperature in regional hyperthermia:
A mathematical approach*,

Authors: Jesús J. Bosque, Gabriel F. Calvo, Víctor M. Pérez-García and María Cruz
Navarro

Dear Dr Jensen

We have examined the second report from the reviewer 1 and submit a revised manuscript accounting for all the minor points addressed. We would like to thank the reviewer for her/his thorough reading of our manuscript and the Editorial Office of Royal Society Open Science for considering our manuscript for publication.

In what follows we detail our responses to the comments and list the changes made to the manuscript. Changes to the manuscript have been highlighted in blue.

Yours sincerely,

Jesús J. Bosque

Department of Mathematics
MOLAB-Mathematical Oncology Laboratory
Universidad de Castilla-La Mancha
Ciudad Real, Spain

Reviewer 1

We would like to thank the reviewer for her/his thorough reading of the manuscript that helps improve the presentation of our work. We now proceed to reply to all the points raised and describe the changes made in the manuscript, which have been highlighted in blue colour in the main text.

Comment: *Figure 1 and 2: The letters A to D are very large compared to the normal text.*

Reply: In both figures, those letters have been reduced from a font size of 30 points to a font size of 23 points.

Comment: *Figure 1: in the caption the reference to the equation is missing (??).*

Reply: The reference to equations 2.1 has now been fixed.

Comment: *p.8. Include the actual reference to the group's website.*

Reply: The sentence has been slightly changed to refer to the Data Accessibility section, at the end of the paper, where we have placed the links to our code hosted both in our group's website and in the Zenodo repository (as requested by the editorial office).

Comment: *p.8.: How were the stochastic fluctuations of blood flow included? (What probabilities)*

Reply: To get closer to the physical reality in which a damaged vasculature in a tumour would be subject to variability in the flow (see for example Dewhirst, M. W., et al. "Microvascular studies on the origins of perfusion-limited hypoxia." The British journal of cancer. Supplement 27 (1996): S247), we added random profiles that reproduce that biological effect. To do so we used a multiplying factor that modifies the flow. This factor consists of a purely sinusoidal part with 30 minute period and a random part which oscillates every 1.5 minutes with a uniform probability centered in zero, being the amplitude variable with the vasculature impairment parameter, ζ . To better convey this feature we have reworded the explanation and added the requested information in the 'Implementation and numerical solution' subsection, page 8, including the new reference to the aforesaid article.

Comment: *p.8: What about the hyperparameters of EqSolver (assumably default?)*

Reply: The function `HTforBrainTumour` is the top level routine in our code's hierarchy. The values of the parameters are set in it, and subsequently passed to the lower-level function `EqSolver`. The default values of the parameters are set to the values that are provided in the table 1 of the manuscript. To facilitate the data flow we grouped the parameters in seven structs associated to a specific category, for instance, parameters related to geometry, parameters related to heat transfer, etc. Therefore, the only variables that are passed from `HTforBrainTumour` to `EqSolver` are these structs that pack all the information and simplify the writing and readability of the code.

To clarify the pipeline that the different functions follow we have added in the 'Implementation and numerical solution' subsection a brief comment related to the exchange of grouped parameters between `HTforBrainTumour` and `EqSolver` (page 8). We have also added to the code a brief `.txt` 'readme' file describing these pipeline.

Comment: *Figure 4/5/6: Although given in the text, a legend in the figure would help to quickly assess which lines are which tissue. In Figure 5 indicate the power level in the figure, too. There is a line under the degree symbol on the y-axis.*

Reply: A legend has been added at the bottom of the figure to help easily identify the correspondence between colours and meaning. Additionally, when different powers are considered in the same figure (i.e. in figure 5 and figure 6), an indication of these powers has been placed on the top of the figures.

Comment: *p.18.l.20: Correct 'Or well' in blue text*

Reply: The typo has now been fixed.

Comment: *Published code: A brief 'readme' file outlining the different functions, their role, and how they should be executed together would help potential users.*

Reply: Following the reviewer instructions we have added a brief .txt 'readme' file that captures the relationships between the functions and may be useful to future users.

This concludes our reply to the reviewer comments.

Sincerely,

The authors